# Potassium nutrient response in the rice-wheat cropping system in different agro-ecozones of Nepal

Roshan Babu Ojha[ID]$^{1}$&*, Shova Shrestha$^{1}$&, Yajna Gajadhar Khadka$^{1}$&, Dinesh Panday[ID]$^{2}$&

1 Soil Science Division, Nepal Agricultural Research Council, Khumaltar, Lalitpur, Nepal, 2 Department of Biosystems Engineering and Soil Science, University of Tennessee-Knoxville, Knoxville, TN, United States of America

& These authors contributed equally to this work.
* roshanbachhan@gmail.com

**Data Availability Statement:** All relevant data are within the paper and its Supporting Information files.

**Funding:** Nepal Agricultural Research Council, Singhadurbar Plaza, Kathmandu, Nepal funded the

## Abstract

Most of the soils of Nepal had a higher potassium (K, expressed as $K_2O$) level inherently. Later in 1976, the Government of Nepal has recommended K fertilizer rate at 30 kg $K_2O$ ha$^{-1}$ in rice-wheat cropping systems. However, those crops began showing K deficiency symptoms in recent decades, which could be due to a large portion of soils with depleted K level or the insufficient input of K fertilizer for crop production. This study explored a limitation of K nutrient in the crops by establishing field trials from 2009–2014 at three agro-ecozones i.e., inner-Terai (2009–2010), high-Hills (2011–2012), and Terai (2012–2014) in Nepal. Seven rates of K fertilizer at 0, 15, 30, 45, 60, 75, and 90 kg $K_2O$ ha$^{-1}$ were replicated four times in a randomized complete block design, where crop yields and yield-attributing parameters of rice-wheat cropping system were recorded. Results revealed that an increase in K rates from 45 to 75 kg $K_2O$ ha$^{-1}$ under inner-Terai and Terai conditions and 45 to 60 kg ha$^{-1}$ under high-Hills conditions produced significantly higher grain yields compared to the recommended K dose. Economically, the optimum rate of K fertilizer should not exceed 68 kg $K_2O$ ha$^{-1}$ for rice in all agro-ecozones, or 73 kg $K_2O$ ha$^{-1}$ for wheat in inner-Terai and 60 kg $K_2O$ ha$^{-1}$ for wheat in high-Hills and Terai. Our findings suggest to increase potassium application in between 1.5 to 2.5 times of the current K fertilizer rate in rice-wheat cropping system of Nepal that need to be tested further in different locations and crop varieties.

## 1 Introduction

The rice (*Oryza sativa* L.)-wheat (*Triticum aestivum* L.) cropping system (RWCS) is the most important cropping system in the subtropical zone of South and Southeast Asia [1]. It covers an estimated 13.6 million ha in the Indo-Gangetic Plains (IGP) of India, Bangladesh, Nepal, and Pakistan [2] and provides a livelihood for hundreds of millions of people that contributes to regional food and nutritional security [3, 4]. Rice-wheat crop rotation is also the most common rotation system in Nepal [5, 6]. In this system, rice is usually cultivated during the warm

project from the regular budget only to conduct the multi-location field trials. The funder had no role in the study design, data collection and analysis, decision to publish, or preparation of the manuscript.

**Competing interests:** The authors have declared that no competing interests exist.

monsoon season (June to September), while wheat is subsequently grown in the cold, humid season (November to February).

Soils such as Ustochrepts, Dystochrepts, and Haplumbrepts are common Inceptisols in Nepal, which are dominant with illite, mica, chlorite, and kaolinite clay minerals [7–9]. Inceptisols are fragile, weakly developed, and susceptible to management [10]. The Inceptisols of Nepal were rich in K due to the presence of mica (a hydrous potassium aluminum silicate mineral) and interlayer fixed K in 2:1 clay minerals that contribute most to the passive K pool, though easy mineralization of mica also contributes to the active K pool [11, 12].

In 1976, a soil test-based fertilizer recommendation was made in Nepal by correlating crop response data with soil test values deeming the importance of economic return per unit of fertilizer use [13]. Due to the K rich soil, a low rate of K (30 kg $K_2O$ ha$^{-1}$) was recommended for rice and wheat to maintain yield. But, during the past 44 years, no update on soil fertilizer rate has been made and farmers use minimal K fertilizer in the field apart from organic sources, predominantly farmyard manure [14, 15]. This resulted a deficiency of K nutrient in field crops overtime. Potassium (K) is one of the major nutrients for crop production, plays a significant role in metabolic reactions in plants by activating a multitude of enzymes and creating a positive effect on plant water stability and deficiency causes, reducing sugar accumulation and decreasing organic acid [16].

Several studies reported a substantial change in K present in mineral parent material K and fertilizer management in cropland on a decennial timescale [17, 18]. Due to mineral dissociation and decreased K balance in soil, Nepal is now experiencing a deficiency of K in field crops [19, 20]. Subsequently, the K consumption rate of Nepal in 2009 was 0.45 kg $K_2O$ ha$^{-1}$, which slightly increased in 2016 to 1.5 kg $K_2O$ ha$^{-1}$; however, this is still far less than the national recommended dose of 30 kg $K_2O$ ha$^{-1}$ [21]. So, we considered 15 kg $K_2O$ ha$^{-1}$ as a basal recommended K dose (BRKD) in the current experiment [22].

For instance, the yield of RWCS in recent decades is decreasing mainly due to a decline in soil fertility [23, 24]. Yields of RWCS are also declining due to the incidence of diseases such as rusts, leaf blight, spot blotch [25], and a change in temperatures in the region during the grain filling period [26]. Many reports have indicated a decreasing trend in soil fertility, including K concentration in soils of Nepal [19, 27]. A recent soil test report from soil testing mobile van program in 9 (out of 75) districts in Nepal revealed that around 33% of total sample tested (n = 1479) had soil K at 10–30 kg $K_2O$ ha$^{-1}$ or even less [28]. Due to farmers' use of farmyard manure, mineralized K from farmyard manure is the primary K input in soil. The current farmyard manure application rate among Nepalese farmers is about 2.5–3 t ha$^{-1}$ annually; however, the inferior quality of manure (N– 0.5 to 0.8%, P– 0.2%, and K– 0.5%) [29] has resulted in a very high farmyard manure recommendation rate (20–30 t ha$^{-1}$), which is far beyond farmers' achievability [30, 31]. Thus, additional K sources are necessary to replenish the mined K from cropping soils.

A proper nutrient management plan is necessary to sustain yield in the long run with incorporation of K fertilizer in the fertility management program. Improved productivity of RWCS in Nepal with increased rates of K fertilizer is also a major concern to keep up with population growth in Nepal, which is predicted to be 36 million by mid-2050 [32, 33]. Because increasing K levels in K deficient soils may increase crop productivity in RWCS, our objective was to investigate the yield response and nutrient response of additional K fertilizer in rice and wheat in three predominant RWCS agro-ecozones in Nepal: the inner-Terai, Terai, and high-Hills.

## 2 Materials and methods

### 2.1 Study area and climate

Nepal is divided into five agro-ecozones (Terai, inner-Terai, mid-Hills, high-Hills, and high-Himalayas). Elevation ranges from 80 meters above sea level (masl) in Terai to 8,848 masl in the high-Himalayas. Cultivation is conducted up to the 4,800 masl high-Hills [34]. The climate in all agro-ecozones is subtropical with warm and wet summer and cold and humid winter [35]. Average annual rainfall ranges from 1000–1800 mm and more than 80% of annual rainfall occurs during the monsoon period, while the winter season receives lesser rainfall than monsoon [35].

The current study was conducted in research stations at the Nepal Agricultural Research Council (NARC), an apex government institution of agriculture research representing different agro-ecozones of Nepal. This study was a regular annual project of the Soil Science Division, Nepal Agricultural Research Council (NARC). All of the multi-location trials within NARC research stations were permitted to conduct field trials by collaborating with their respective research stations. The institutional review board to accept this study was the proposal evaluation committee of NARC, Singhadurbar plaza, Kathmandu, Nepal. The first research site was established in 2009–2011 at the National Maize Research Program, Rampur, Chitwan, which represents the inner-Terai agro-ecozone situated at the coordinates 27˚39' N, 84˚20' E and an elevation of 187 masl. The second research site was established in 2010–2011 at Hill Research Station, Kabre, Dolakha, representing the high-Hills agro-ecozone with the coordinates 27˚38' N, 86˚9' E and an elevation of 1820 masl. The third research site was established in 2012–2014 at the Regional Agriculture Research Station, Parwanipur, Bara, which represents the Terai agro-ecozones with the coordinates 27˚4' N, 84˚55' E and an elevation of 96 masl. The crop cycle in each research site was rice followed by wheat at all years during the research period.

Weather data was collected in the research station. Symon's rain gauze was used to measure precipitation and a maximum-minimum thermometer was used to record the temperature. Daily weather parameters were recorded that were then averaged over the months and years. During the research period, the Kabre site received 2370 mm annual precipitation with the maximum rainfall (790 mm) in August and the minimum (0 mm) rainfall from December to February. The annual temperature ranged from 26˚C (in May and June) to 3˚C (in January). The average maximum and minimum temperatures were 22˚C and 12˚C, respectively, for the region. Similarly, the Rampur site received 2290 mm annual precipitation with the maximum rainfall (697 mm) in August and the minimum rainfall (0 mm) in November and January. The annual temperature ranged from 37˚C (in April) to 10˚C (in December and January). The average maximum and minimum temperatures were 31˚C and 21˚C, respectively. The Parwanipur site received 1423 mm annual precipitation with the maximum rainfall (311 mm) in August and the minimum rainfall (0 mm) from November to January and in April. The annual temperature ranged from 37.5˚C (in April) to 10˚C (in January). The average maximum and minimum temperatures were 22˚C and 12˚C, respectively.

### 2.2 Soil

Baseline soil sampling from the top 20 cm depth were collected and analyzed for soil texture, pH, organic carbon (OC), total N, available P, and available K concentrations (Table 1). Soil samples from the top 20 cm were again collected from experimental plots in 2012 (Kabre site), 2010 (Rampur site), and 2013 (Parwanipur site) to observe changes in pH, OC, total N, available P, and available K concentrations before the planting of the main crop. After the harvest

**Table 1. Initial soil parameter test results, safe nutrient range, and major K minerals of the research sites (safe limit/ medium range values adopted from Soil Science Division [36]).**

| Location | Safe limit/ medium range | Current result | Major K mineral [13] |
|---|---|---|---|
| **High hills** | | | |
| pH | 6.50–7.50 | 6.20 | Mica |
| OC, % | 2.50–5.00 | 0.75 | |
| Total N, % | 0.10–0.20 | 0.07 | |
| Available $P_2O_5$[‡], kg ha[-1] | 13.00–25.00 | 47.52 (107.30) | |
| Available $K_2O$[‡], kg ha[-1] | 49.00–125.00 | 100.83 (121.00) | |
| Soil texture | | Sandy loam | |
| **Terai** | | | |
| pH | 6.50–7.50 | 5.8 | Mica |
| OC, % | 1.40–2.81 | 1.27 | |
| Total N, % | 0.10–0.20 | 0.14 | |
| Available $P_2O_5$, kg ha[-1] | 13.00–25.00 | 52.44 (118.00) | |
| Available $K_2O$, kg ha[-1] | 49.00–125.00 | 50.83 (61.00) | |
| Soil texture | | Sandy loam | |
| **Inner-Terai** | | | |
| pH | 6.5–7.5 | 6.20 | Mica |
| OC, % | 0.84–1.68 | 0.63 | |
| Total N, % | 0.07–0.15 | 0.11 | |
| Available $P_2O_5$, kg ha[-1] | 13.00–25.00 | 21.33 (48.00) | |
| Available $K_2O$, kg ha[-1] | 49.00–125.00 | 137.50 (165.00) | |
| Soil texture | | Silty clay | |

[‡] Conversion factor: $P_2O_5$ = 2.25 x P and $K_2O$ = 1.20 x K

of the crops, three soil samples from the top 20 cm of a plot were composited by mixing and removing gravel, roots, undecomposed materials, and debris. The collected soil samples were then air-dried, ground to break up aggregates, and run through a 2 mm sieve before being stored and subjected to lab analysis.

Soil chemical analysis followed a protocol developed by the SSD to analyze the soil. Soil pH was determined in a 1:2 soil water ratio using electric pH electrodes. Soil organic carbon was determined by the wet digestion method as described by the Walkley-Black method [37]. The Kjeldahl method was used to determine total N in which soil is digested in concentrated sulphuric acid [38], while available P was determined in a spectrophotometer as described by a modified Olsen method [39]. Available K from the soil was extracted in a 0.1 M ammonium acetate solution and determined using a flame photometer [40].

## 2.3 Experimental design and treatment

At all three sites, the experimental design was laid out in a randomized complete block design with four replications. Treatment included seven rates of K fertilizer as 0, 15, 30, 45, 60, 75 and 90 kg $K_2O$ ha[-1]. We considered 15 kg $K_2O$ ha[-1] as basal recommended K dose (BRKD). The N and P fertilizers were applied at a rate of 100 kg N ha[-1] and 30 kg $P_2O_5$ ha[-1] in each plot [13]. Half rates of N, full rates of P, and full rates of K were applied as basal at the time of transplanting and the remaining half rate of N was applied at the tillering stage. All of the fertilizers were distributed and ploughed immediately during basal dose application in both crops.

In this study, urea, triple superphosphate (TSP), and muriate of potash (MOP) were sources of N, P, and K, containing 46% N, 46% $P_2O_5$, and 60% $K_2O$, respectively. No organic fertilizer

sources were used in the research plots. All of these fertilizers were sourced from Salt Trading Corporation Limited, Kathmandu (coordinates—27˚ 40' 39.77"N, 85˚ 21' 12.05" E; fertilizer source origin–India; appearance of urea–white, crystals form, and solid state; TSP–light brownish, granular form, and solid state; MOP–reddish, mixed crystal and powder form, solid state). Similarly, rice and wheat seeds were sourced from the Botany Division of NARC and were the recommended varieties for the regions. The WK 1204 variety of wheat and Ram Dhan variety of rice were selected for the high-Hill region and the Gautam variety of wheat and Hardinath variety of rice were selected for the inner-Terai and Terai regions. Wheat seeds were directly sown in the field while 21 day-old rice seedlings were transplanted in the puddled field in a row-to-row spacing of 20 cm and hill-to-hill spacing of 20 cm.

## 2.4 Statistical analysis

Yield and yield parameters of rice and wheat were collected during the growing seasons. Length parameters (plant height and panicle length) were measured by using scale and weight parameters (grain yield, straw yield, biomass, and thousand-grain weight) were measured. Tiller number was recorded with manual counting. Relative yield (RY) was calculated by subtracting control yield from treatment yield (delta yield), divided by control yield, and expressed as a percentage.

$$RY\ (\%) = \frac{delta\ yield}{control\ yield} x100 \tag{Eq 1}$$

The maximum physical K ratew was calculated by equating the first-order derivative of the yield response function with zero and the economic optimum K rate was calculated by equating the first-order derivative of the yield response function with the price ratio [41, 42].

$$\text{For maximum physical K rate, } \frac{\delta y}{\delta x} = \mathbf{0} \tag{Eq 2}$$

$$\text{For economic optimum K rate, } \frac{\delta y}{\delta x} = \frac{Py}{Px} \tag{Eq 3}$$

Where, δy = yield response function, δx = derivative for x, Py = unit price of fertilizer, Px = unit price of grain yield.

The assumption of analysis of variance (ANOVA) was tested and means were separated at alpha 0.05 level of significance using Tukey's test in R-studio software (v. 1.3.1056). Time (in years) and location were not combined to analyze as fixed variables and were treated individually. A correlation test was carried out between different parameters at alpha 0.05 level of significance.

## 3 Results

### 3.1 Soil chemical properties

In a year, changes in soil chemical properties due to the addition of different rates of K fertilizer were measured after harvest of the first year's crop (in different years: Kabre in 2012, Rampur in 2010 and Parwanipur in 2013) at different sites and their values are presented in S1 Table in S1 File. There were no significant changes in pH, OC, N, and P (except at the Rampur site) in addition to K at different rates. Soil K concentration increased with addition of K rates at the Kabre site compared to the control. Only K rates at 60 kg $K_2O$ ha$^{-1}$ or higher increased soil K concentration compared to the control at the Rampur site (S1 Table in S1 File).

## 3.2 Yield response

**3.2.1 Inner Terai (Rampur).** Thousand-grain weight, grain yield, and straw yield of rice and wheat increased with the addition of K fertilizer in the inner-Terai agro-ecozone (Table 2). Results showed an increased demand for K fertilizer to obtain a significantly higher yield of rice than the BRKD (15 kg $K_2O$ ha$^{-1}$). A potassium fertilizer rate of at least 45 kg $K_2O$ ha$^{-1}$ or higher produced significantly greater yields, with a trend of $r^2 = 0.69$, $P = 0.122$ in 2009, $r^2 = 0.55$, $P = 0.26$ in 2010 and $r^2 = 0.94$, $P = 0.006$ in 2011. Similarly, a significantly higher rice straw yield was obtained from 75 kg $K_2O$ ha$^{-1}$ than the BRKD in the first and third years. However, rice yield declined up to 30–33% in the second year (Table 2).

Thousand-grain weight (TGW) of rice was significantly increased compared to the BRKD in the first and third years from 75 and 60 kg $K_2O$ ha$^{-1}$, but no significant increment was observed in the second year (Table 2). Plant height was affected by K rate 75 kg $K_2O$ ha$^{-1}$ and significantly higher than the BRKD, but panicle length and tiller number did not differ significantly with the BRKD (S2 Table in S1 File). All of these yield-attributing traits were positively correlated with yield, but a strong positive correlation ($r^2 = 0.87$, $p < 0.0001$) was reported with plant height (S5 Table in S1 File).

Wheat grain and straw yield significantly increased compared to the BRKD with an additional level of K (Table 3). The significant yield of wheat increased from 30 to 75 kg $K_2O$ ha$^{-1}$ in the first and second year, however, a 38% decline in grain yield and 20% in straw yield was observed over the three years (Table 3).

Thousand-grain yield was significantly increased at 60 kg $K_2O$ ha$^{-1}$ compared to the BRKD. Plant height was significantly higher than the BRKD, with the highest plant height reported from the 90 kg $K_2O$ ha$^{-1}$ dose in the first year and the 75 kg $K_2O$ ha$^{-1}$ dose in the second year (Table 3). There was no significant increment in panicle length and tiller number with the BRKD and no significant increment in tiller number with the control in the second and third years (S4 Table in S1 File). The highest positive correlation was observed between wheat grain yield and plant height ($r^2 = 0.82$, $p = 0.000$) and panicle length ($r^2 = 0.74$, $p = 0.000$) (S5 Table in S1 File).

**3.2.2 High hills (Kabre).** The additional K rate had no significant response over the BRKD in straw yield and grain yield of wheat in both years and of rice in the first year

**Table 2. Analysis of variance (ANOVA) results with means for thousand-grain weight, grain yield, and straw yield of rice as affected by different rates of potassium fertilizer in a rice-wheat cropping system at Rampur, Chitwan, Nepal over three consecutive years.**

| Treatment,kg $K_2O$ ha$^{-1}$ | Thousand-grain weight, g | | | Grain yield, kg ha$^{-1}$ | | | Straw yield, kg ha$^{-1}$ | | |
|---|---|---|---|---|---|---|---|---|---|
| | **2009** | **2010** | **2011** | **2009** | **2010** | **2011** | **2009** | **2010** | **2011** |
| 0 | 19.0c[†] | 21.9 | 19.2b | 2210.0c | 1978.0c | 2102.0c | 2762.0c | 2419.0b | 2392.0c |
| 15 | 19.5b | 21.7 | 19.6ab | 4439.0b | 3259.0b | 3010.0b | 5256.0b | 4056.0a | 4032.0b |
| 30 | 20.0ab | 20.1 | 19.9ab | 4818.0ab | 3609.0ab | 3219.0ab | 6250.0ab | 3903.0a | 4541.0ab |
| 45 | 19.8b | 20.6 | 19.8ab | 4960.0a | 3502.0ab | 3129.0ab | 6037.0ab | 3944.0a | 4538.0ab |
| 60 | 20.0ab | 22.1 | 20.3a | 5161.0a | 3875.0a | 3289.0ab | 6002.0b | 4376.0a | 4335.0ab |
| 75 | 20.5a | 21.9 | 20.3a | 5241.0a | 3641.0ab | 3555.0a | 7067.0a | 4125.0a | 4862.0a |
| 90 | 19.6b | 22.2 | 20.0a | 4890.0a | 3576.0ab | 3569.0a | 6188.0ab | 3861.0a | 4496.0ab |
| Significance | *** | NS | * | *** | *** | *** | *** | ** | *** |

[†]Means in a column followed by the same lowercase letter are not significantly different.

*$P < 0.05$,

**$P < 0.01$,

***$P < 0.001$, and NS = not significant.

**Table 3. ANOVA results with means for thousand-grain weight, grain yield, and straw yield of wheat as affected by different rates of potassium fertilizer in a rice-wheat cropping system at Rampur, Chitwan, Nepal over three consecutive years.**

| Treatment, kg K₂O ha⁻¹ | Thousand-grain weight, g | | | Grain yield, kg ha⁻¹ | | | Straw yield, kg ha⁻¹ | | |
|---|---|---|---|---|---|---|---|---|---|
| | 2009 | 2010 | 2011 | 2009 | 2010 | 2011 | 2009 | 2010 | 2011 |
| 0 | 42.5cd† | 25.6f | 26.4d | 961.0d | 352.0d | 533.0d | 2663.0c | 1005.0d | 1674.0c |
| 15 | 40.7d | 31.2e | 32.1c | 1930.0c | 895.0c | 1844.0c | 6001.0b | 1705.0d | 4380.0b |
| 30 | 44.5bcd | 39.8b | 35.8b | 3504.0b | 1843.0b | 2289.0b | 6924.0ab | 3054.0bc | 4843.0ab |
| 45 | 45.5bc | 38.1c | 37.7ab | 4025.0ab | 1633.0b | 2882.0a | 7066.0ab | 2734.0c | 5093.0ab |
| 60 | 51.2a | 34.4d | 39.3a | 4190.0ab | 2442.0a | 2761.0a | 7102.0ab | 3796.0ab | 5840.0a |
| 75 | 48.5ab | 43.5a | 35.7b | 4529.0a | 2562.0a | 2628.0ab | 7137.0ab | 3888.0a | 5591.0a |
| 90 | 53.0a | 37.5c | 37.1ab | 4374.0a | 2288.0a | 2853.0a | 7244.0a | 3579.0ab | 5520.0a |
| Significance | *** | *** | *** | *** | *** | *** | *** | *** | *** |

†Means in a column followed by the same lowercase letter are not significantly different.

***$P < 0.001$ and NS = not significant.

(Table 4). Rice grain and straw yield were significantly increased in the second year of potassium application with 30 kg K₂O ha⁻¹. Over the two years, grain yield of rice increased by 10%, but straw yield declined by 22%.

Thousand-grain weight of rice did not significantly increase in either the first or second year (Table 4). Plant height of rice significantly increased at the 45 kg K₂O ha⁻¹ application in the second year, but tiller number and panicle length in rice, and plant height and tiller number in wheat were not significantly different compared with the BRKD (S6 Table in S1 File). Panicle length of wheat was significantly increased at 60 kg K₂O ha⁻¹ compared to the BRKD. A significant correlation of grain yield was observed with plant height of rice ($r^2 = 0.81$, $p = 0.000$), plant height of wheat ($r^2 = 0.68$, $p = 0.000$), and tiller number ($r^2 = 0.75$, $p = 0.000$) (S7 and S8 Tables in S1 File).

**3.2.3 Terai (Parwanipur).** Grain yield and straw yield significantly increased compared to the control but not with the BRKD in rice (Table 5). However, wheat exhibited a significant response of additional K fertilizer from 60 Kg K₂O ha⁻¹. Over the two years, rice yield

**Table 4. ANOVA results with means for thousand-grain weight, grain yield, and straw yield of rice and wheat as affected by different rates of potassium fertilizer in a rice-wheat cropping system at Kabre, Dolakha, Nepal over two consecutive years.**

| Treatment, kg K₂O ha⁻¹ | Rice | | | | | | Wheat (2011) | | |
|---|---|---|---|---|---|---|---|---|---|
| | Thousand-grain weight, g | | Grain yield, kg ha⁻¹ | | Straw yield, kg ha⁻¹ | | Thousand-grain weight, | Grain yield, | Straw yield, |
| | 2010 | 2011 | 2010 | 2011 | 2010 | 2011 | g | kg ha⁻¹ | kg ha⁻¹ |
| 0 | 19.12 | 20.1 | 1783.0b† | 1941.0c | 3870.0b | 2815.0c | 51.9 | 890.0b | 3016.0b |
| 15 | 17.93 | 18.6 | 2937.0a | 2958.0b | 8508.0a | 5181.0b | 54.6 | 2416.0a | 5831.0a |
| 30 | 18.22 | 19.1 | 3408.0a | 3254.0a | 9266.0a | 6154.0ab | 55.4 | 2422.0a | 6988.0a |
| 45 | 18.53 | 19.2 | 3272.0a | 3187.0ab | 8846.0a | 6129.0ab | 54.1 | 2461.0a | 5994.0a |
| 60 | 18.05 | 18.7 | 3017.0a | 3198.0ab | 9043.0a | 6092.0ab | 51.6 | 2634.0a | 6923.0a |
| 75 | 18.03 | 18.7 | 3325.0a | 3297.0a | 9053.0a | 6624.0a | 53.6 | 2215.0a | 6236.0a |
| 90 | 18.4 | 19.4 | 3459.0a | 3256.0a | 8367.0a | 6019.0ab | 53.3 | 2607.0a | 6111.0a |
| Significance | NS | NS | *** | *** | *** | *** | NS | *** | ** |

†Means in a column followed by the same lowercase letter are not significantly different.

** $P < 0.01$,

***$P < 0.001$, and NS = not significant

**Table 5. ANOVA results with means for thousand-grain weight, grain yield, and straw yield of rice and wheat as affected by different rates of potassium fertilizer in a rice-wheat cropping system at Parwanipur, Bara, Nepal.**

| Treatment, kg $K_2O$ ha$^{-1}$ | Rice | | | | | | Wheat | | | | | |
|---|---|---|---|---|---|---|---|---|---|---|---|---|
| | Thousand-grain weight, g | | Grain yield, kg ha$^{-1}$ | | Straw yield, kg ha$^{-1}$ | | Thousand-grain weight, g | | Grain yield, kg ha$^{-1}$ | | Straw yield, kg ha$^{-1}$ | |
| | 2013 | 2014 | 2013 | 2014 | 2013 | 2014 | 2013 | 2014 | 2013 | 2014 | 2013 | 2014 |
| 0 | 19.9 | 22.6 | 1507.0b$^†$ | 1551.0c | 401.0b | 3953.0b | 41.7b | 45.5b | 761.0b | 1029.0d | 1196.0d | 1282.0d |
| 15 | 22.1 | 22.1 | 2287.0a | 2993.0ab | 6823.0a | 6691.0a | 43.3b | 46.3b | 2474.0a | 2452.0c | 3241.0c | 2746.0c |
| 30 | 21.1 | 21.6 | 2423.0a | 2997.0ab | 6351.0a | 7492.0a | 45.9a | 47.2ab | 2614.0a | 2777.0bc | 3472.0bc | 3170.0b |
| 45 | 21.5 | 23.2 | 2095.0a | 2344.0b | 6702.0a | 6018.0a | 46.3a | 44.8b | 2475.0a | 2829.0abc | 3412.0bc | 3211.0b |
| 60 | 22.5 | 23.4 | 2623.0a | 3333.0a | 6707.0a | 7104.0a | 47.7a | 46.9b | 2767.0a | 3310.0a | 3770.0a | 3776.0a |
| 75 | 22.9 | 23.1 | 2092.0a | 2772.0ab | 6827.0a | 6778.0a | 47.4a | 47.3ab | 2633.0a | 3055.0ab | 3655.0ab | 3627.0a |
| 90 | 23.3 | 22.9 | 2159.0a | 3244.0a | 6917.0a | 7160.0a | 47.3a | 50.3a | 2556.0a | 3038.0ab | 3865.0a | 3431.0ab |
| Significance | NS | NS | ** | *** | *** | *** | *** | * | *** | *** | *** | *** |

$^†$Means in a column followed by the same lowercase letter are not significantly different.

$^*P < 0.05$,

$^{**}P < 0.01$,

$^{***}P < 0.001$, and NS = not significant.

increased up to 30% and incremental highest yield over control treatment was reported up to 80%. This incremental yield of rice was achieved up to 60 Kg $K_2O$ ha$^{-1}$ but declined above this. The addition of K did not increase the thousand-grain weight of rice in either year. A 33% increase in wheat grain yield was reported from 60 kg $K_2O$ ha$^{-1}$ compared to the BRKD. Wheat straw yield was similar (3.7 t ha$^{-1}$) at 60 kg $K_2O$ ha$^{-1}$ in both years.

The thousand-grain yield of wheat was significantly higher from 30 kg $K_2O$ ha$^{-1}$ in the first year and 90 Kg $K_2O$ ha$^{-1}$ in the second year compared to the control (Table 5). In both crops yield attribute traits other than tiller number (i.e., plant height and panicle length) significantly increased compared to the control but were similar to the BRKD. Significantly, the highest tiller number of both rice (260 m$^{-2}$) and wheat (312 m$^{-2}$) was obtained from 90 kg $K_2O$ ha$^{-1}$ (S9 Table in S1 File). Strong positive significant correlation exists between yield and yield-attributing traits (plant height and panicle length) of rice ($r^2 = 0.82$, $p < 0.0001$) and wheat ($r^2 = 0.76$, $p = 0.000$), though a weak non-significant correlation ($r^2 = 0.32$, $p = 0.87$) exists with thousand-grain weight (S10 and S11 Tables in S1 File).

### 3.3 Nutrient response

**3.3.1 Physical maximum and economic optimum rates.** Physical maximum and economic optimum rates of K for rice and wheat at all agro ecozones were near values (Table 6). The rate of K for physical maximum production was highest for wheat in inner-Terai at 4.8 times the recommended rate, and was the least for rice in Terai at 3.4 times the recommended rate. Similarly, the K requirement for wheat should be increased by 4.7 times in inner-Terai and 3.9 times in the high-Hills and Terai.

**3.3.2 Expected yield and relative yield over residual fertility yield.** Additional K rates increased the relative yield of rice and wheat over soil inherent/residual fertility (yield obtained from the control plot) with incremental expected yield (Fig 1). The residual fertility rice yield is around 2 t ha$^{-1}$ in all agro-ecozones. The increased K rate was expected to increase the yield of rice maximally by 1.8, 0.9, and 1.3 t ha$^{-1}$ in inner-Terai, Terai, and high-Hills, respectively. Similarly, the residual fertility yield of wheat was 0.6 t ha$^{-1}$ in inner-Terai and around 1 t ha$^{-1}$ in Terai and high-Hills. The increased K rate was expected to increase the yield of wheat

**Table 6. Physical maximum rate and economic optimum rate of potassium fertilizer recommended for the rice-wheat cropping system in different agro-ecozones of Nepal.**

| Agro-ecozones | Crop | Nutrient response curve function[†] | Physical maximum rate | Economic optimum rate |
|---|---|---|---|---|
| | | | kg K$_2$0 ha$^{-1}$ | |
| High-Hills (Dolakha) | Rice | $y = -0.4627x^2 + 58.401x + 2397.1$ | 68.0 | 67.0 |
| | Wheat | $y = -0.5017x^2 + 72.826x + 634.0$ | 60.0 | 58.0 |
| Inner-Terai (Chitwan) | Rice | $y = -0.2535x^2 + 30.839x + 1812.2$ | 63.0 | 57.0 |
| | Wheat | $y = -0.4946x^2 + 60.738x + 1197.4$ | 73.0 | 71.0 |
| Terai (Bara) | Rice | $y = -0.3245x^2 + 41.085x + 2121.3$ | 51.0 | 50.0 |
| | Wheat | $y = -0.3983x^2 + 47.665x + 1255.3$ | 61.0 | 59.0 |

[†]y = yield obtained (dependent variable) and x = fertilizer dose (independent variable). The nutrient response curve function was derived based on the averaged data over the years of the respective locations.

maximally by 2.6, 1.8, and 1.4 t ha$^{-1}$ in inner-Terai, Terai, and high-Hills, respectively. The response of K was high in inner-Terai due to the low residual potassium level.

## 4 Discussion

Increased yields of rice and wheat in the inner Terai and Terai regions of Nepal with K fertilizer application rates of 50 to 60 kg K$_2$O ha$^{-1}$ (Tables 2, 3, and 5) make it evident that these regions have low availability of nutrients due to sandy soil, high rainfall intensity, and frequent leaching loss of K [20]. Other findings also reported low availability of K in these regions [5, 43, 44]. The soils of these regions are generally characterized as fertile soil, which is made up of recent alluvial deposits, mostly fine sand and silt with light to medium texture, with the Terai region of Nepal considered the basket of grain [45, 46]. Crop cultivation in the Terai region began after the clearing the forest in 1927 [47], and robust agricultural production was observed with the alluvial deposit, medium-textured, forest soils. In later decades, however, low organic input, nutrient removal due to crop harvesting, heavy tillage work, continuous erosion, and poor crop and land management resulted in a low nutrient reserve [48, 49]. Hence, K, an active component of the nutrient cycle, is one of those nutrients impacted by low nutrient reserve.

In high-Hills, increased crop yields were recorded with additional K rates in the second year starting at 30 kg K$_2$O ha$^{-1}$ up to 70 kg K$_2$O ha$^{-1}$ (Table 4). Soil in the region is low in OC content, with a high erosion rate and sandy loam texture with acidic soil pH [50, 51]. Additionally, the mid-and-high-Hills are characterized by an erosion rate of 32–38 t ha$^{-1}$ yr$^{-1}$ [52]. Carson [53] reported that a 1 mm loss of topsoil was estimated to cause a loss of 10 kg N ha$^{-1}$, 15.7 kg P$_2$O$_5$ ha$^{-1}$, and 22.5 kg K$_2$O ha$^{-1}$. Bijay-Singh and Vashishta [54] reported that to produce rice and wheat at 8 to 10 t ha$^{-1}$, it can remove around 250 kg K$_2$O ha$^{-1}$ from the soil. In contrast, rice-growing farmers in the mid-Hill regions have benefited from the accumulation of eroded sediments (for example, mica), a source of exchangeable K [55–57]. Additionally, soil pH is a limiting factor for K availability in the high-Hills (S1 Table in S1 File). The response of additional K fertilizer compared to the BRKD did not significantly increase in wheat and rice yields (Table 4) because the acidic pH (4.4) might restrict the availability of K, as microbial nutrient transformation is restricted below a soil pH of 5.5 [58].

Comparing the K fertilizer consumption under RWCS in the IGP region, Nepal uses less K fertilizer in rice (0.9 kg K$_2$O ha$^{-1}$) and wheat (2.0 kg K$_2$O ha$^{-1}$) than its neighbors. India uses 9.4 kg K$_2$O ha$^{-1}$ in rice and 4.6 kg K$_2$O ha$^{-1}$ in wheat, whereas Bangladesh uses 9.9 kg K$_2$O ha$^{-1}$ in rice and 7.9 kg K$_2$O ha$^{-1}$ in wheat [2]. In Asia, China has the highest K use rate in both rice

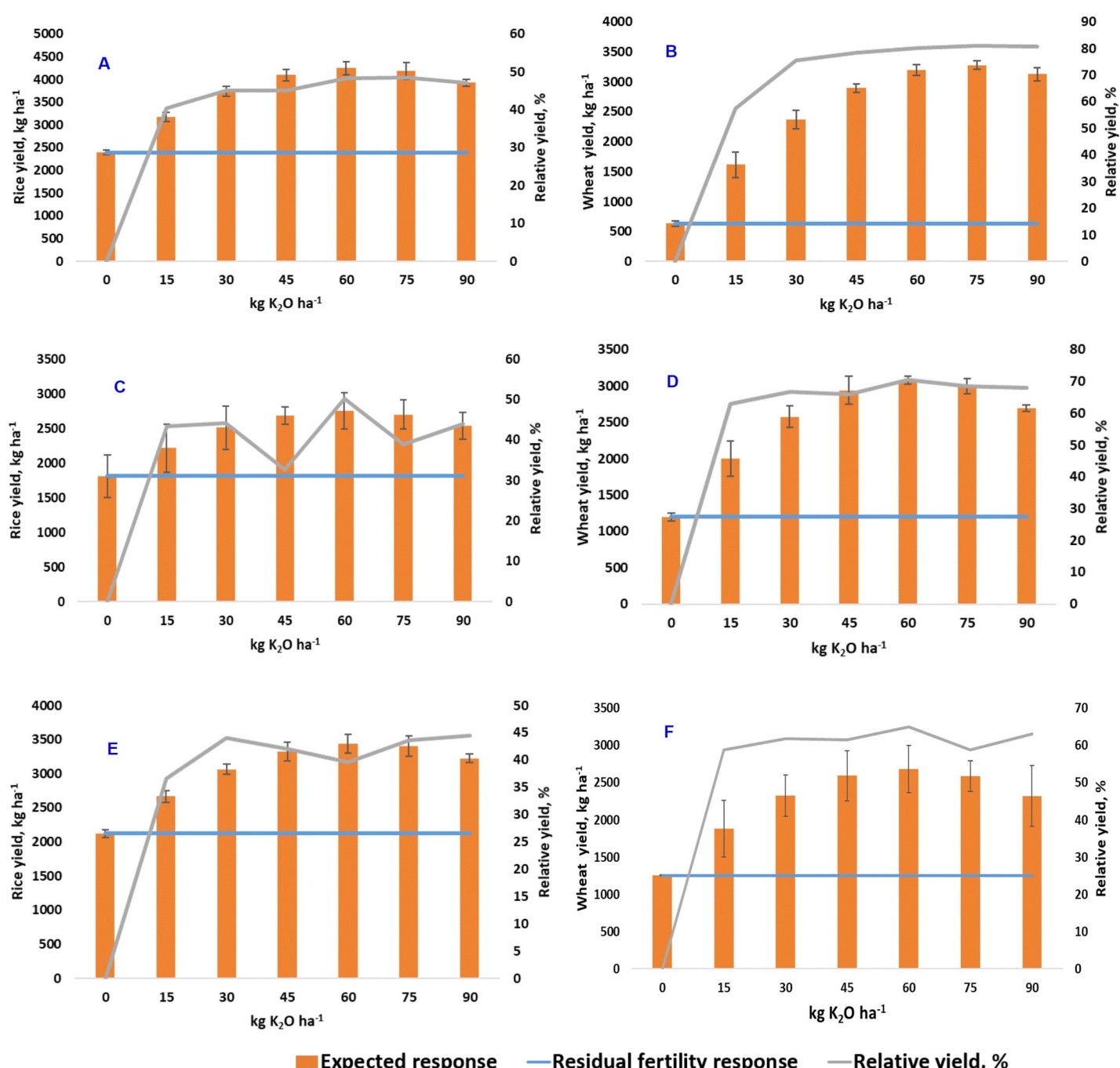

**Fig 1.** Aggregated data analysis to compare the different levels of potassium application on expected yield (represented by the bar graph) and relative yield (represented by the green line graph) over residual fertility yield (represented by the straight blue line) in rice (Fig 1A, 1C and 1E) and wheat (Fig 1B, 1D and 1F) in inner-Terai (1A, 1B), Terai (1C, 1D), and high-Hills (1E, 1F).

(33.2 kg $K_2O$ ha$^{-1}$) and wheat (26.6 kg $K_2O$ ha$^{-1}$). A long-term rice-wheat fertility experiment conducted in Bhairahawa, Nepal showed that annual K balance was negative and that K exhibited a significant response in rice [5]. The increase in yield of both rice and wheat was evident in all three agro-eco zones of Nepal.

Nutrient response functions showed that the physical and economic limits of K rate were more or less equal at all locations. The K requirement of rice should be increased by 4.4, 3.8, and 3.3 times BRKD in high-Hills, inner-Terai, and Terai, respectively. In the current study, the economic optima of K ranged from 50 to 67 kg $K_2O$ $ha^{-1}$ in rice and 58 to 71 kg $K_2O$ $ha^{-1}$ in wheat in different locations (Table 6). The availability of K in soil is proportional to the added K fertilizer [59], but a higher rate of K addition results in fixation rather than availability [56]. As potassium is a so-called luxury consumption nutrient, additional K rates can sometimes result in loss [60]. In addition, K is a mobile element and additional K results in leaching under flooded conditions in the case of rice cultivation [58]. Thus, the current study was unable to attain a more economic yield with 90 kg $K_2O$ $ha^{-1}$, meaning that it is not necessary to apply more than 70 kg $K_2O$ $ha^{-1}$, starting from 50 kg $K_2O$ $ha^{-1}$ depending on the crop type and agro-eco zone.

At all locations, additional K rates increased relative yield of rice and wheat. Rice yield was estimated to increase up to 3 to 4.5 t $ha^{-1}$ (an increase of 40 to 50%) and wheat yield was estimated to increase up to 2.5 to 3 t $ha^{-1}$ (an increase of 60 to 80%) (Fig 1). This different K response in rice and wheat is observed due to the K limitation in the soil and the availability of K after K fertilizer application. The soils of the studied region were high in mica content, and in soils containing high mica, even 1 to 2% of total K is enough. However, continual crop removal of K and restricted K application in soil for long periods resulted in the weathering of mica into biotite or vermiculite, an avenue for K limitation [61]. Similarly, the exchange of K is possible between available and fixed pools. More than 50% of total K availability in RWCS is obtained from fixed K pools that may further deplete the K reserve from the soil [44]. Thus, it is imperative to apply K fertilizer to maintain K levels and soil fertility in the cropland.

Soil N and P are also known to affect K availability and uptake by plants. Application of N and P are reported to have resulted in a 145% increase in K uptake compared to a control [62] and Tiwari et al. [63] reported that response to K application in rice increased with increasing rates of N application. Under nitrogenous fertilization, interlayer K can also be replaced by $NH_4^+$, as both have a nearly similar ionic size, influencing availability. The current study reported that there were no differences in soil total N after a year's addition of K fertilizer, suggesting no adverse effects on total N due to treatment application. However, our study limits that we did not observe the effects of different rates of K fertilizer on uptake of nutrients from the soil by plant roots for grain yield and biomass production.

Nepalese farmers are aware of the importance of chemical fertilizers in crop production and the application of chemical fertilizers is gradually increasing. Most farmers use only N-related fertilizers, which may allow them to reach potential yields. However, this can increase the cost of cultivation in the short term and cause reduction in soil quality and productivity in the long term. Because a continuous and increased application of N fertilizer is not enough to replenish lost plant nutrients and maintain soil productivity [27, 64], a balanced fertilization is necessary to increase the productivity of RCWS.

The current study suggests to increase the current K rate of 30 $K_2O$ $ha^{-1}$ to 60 to 70 kg $K_2O$ $ha^{-1}$ in different agro-eco zones. This field trial uses a single variety of rice and wheat, which was suitable to the soils of research sites, while the K rate might differ for hybrid cultivars of rice and wheat and different soil types in different agro-ecozones. So, a series of field experiments in other agro-ecozones are suggested using hybrid and improved varieties of the crops in different soil types and crop rotations.

## 5 Conclusion

Improvements in crop yield and soil nutrient response with the addition of K fertilizer in RWCS at three agro-ecozones of Nepal suggests that the current K fertilizer rate (30 kg $K_2O$

ha$^{-1}$) should be increased about 1.5 to 2 times for rice and 2 to 2.5 times for wheat to achieve optimum economic production. These recommendations are made for the rice and wheat varieties with a yield potential of 2 to 4 t ha$^{-1}$. The K rate suggested for rice is 67, 57, and 50 kg K$_2$O ha$^{-1}$ and for wheat is 58, 71, and 59 kg K$_2$O ha$^{-1}$ for high-hills, inner-Terai, and Terai, respectively. Fertilizer recommendation is a dynamic and continuous process that largely depends on soil type, crop response, inherent fertility, grain to fertilizer price ratio, and environmental factors, and recommendations should be revised regularly over time while considering these factors. Potassium, an integral crop nutrient component that contributes to soil fertility and optimum crop production, should be applied at the recommended rate in a balanced way by appraising available K sources.

## Supporting information

**S1 File.**
(DOCX)

## Acknowledgments

The authors would like to thank Shambhu Raut for assisting in field data collection along with all of the technical and administrative staff of the Regional Agricultural Research Station, Parwanipur, Bara; the National Maize Research Program, Rampur, Chitwan; Hill Research Station, Kabre, Dolakha; and the Soil Science Division, Khumaltar, Lalitpur. We also thank Bikesh Twanabasu for technical support and Ian Rogers for English editing during the preparation of the manuscript. Finally, we would like to thank anonymous reviewers and an academic editor for their valuable comments and suggestions which helped us in improving this paper.

## Author Contributions

**Conceptualization:** Yajna Gajadhar Khadka.

**Data curation:** Roshan Babu Ojha, Shova Shrestha.

**Formal analysis:** Roshan Babu Ojha.

**Investigation:** Yajna Gajadhar Khadka.

**Methodology:** Roshan Babu Ojha, Yajna Gajadhar Khadka.

**Resources:** Shova Shrestha, Dinesh Panday.

**Validation:** Roshan Babu Ojha, Dinesh Panday.

**Visualization:** Roshan Babu Ojha.

**Writing – original draft:** Roshan Babu Ojha, Dinesh Panday.

**Writing – review & editing:** Roshan Babu Ojha, Shova Shrestha, Yajna Gajadhar Khadka, Dinesh Panday.

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
