## [Decision Letter · Decision Letter 0]

23 Jul 2020

PONE-D-20-12836

Revised potassium nutrition in rice-wheat cropping system of Nepal

PLOS ONE

Dear Dr. Ojha,

Thank you for submitting your manuscript to PLOS ONE. After careful consideration, we feel that it has merit but does not fully meet PLOS ONE’s publication criteria as it currently stands. Therefore, we invite you to submit a revised version of the manuscript that addresses the points raised during the review process.

We look forward to receiving your revised manuscript.

Kind regards,

Rafiq Islam, Ph.D.

Academic Editor

PLOS ONE

Journal Requirements:

2. We understand that you purchased seeds and fertilizers from local markets for this study. In your Methods section, please provide additional regarding the source of this material. Please provide the geographic coordinates and names of the purchase locations (e.g., stores, markets), if available, as well as any further details about the purchased items (e.g., lot number, source origin, description of appearance) to ensure reproducibility of the analyses.

3. We note that [Figure(s) 1] in your submission contain [map/satellite] images which may be copyrighted. All PLOS content is published under the Creative Commons Attribution License (CC BY 4.0), which means that the manuscript, images, and Supporting Information files will be freely available online, and any third party is permitted to access, download, copy, distribute, and use these materials in any way, even commercially, with proper attribution. For these reasons, we cannot publish previously copyrighted maps or satellite images created using proprietary data, such as Google software (Google Maps, Street View, and Earth). For more information, see our copyright guidelines: http://journals.plos.org/plosone/s/licenses-and-copyright.

1.    You may seek permission from the original copyright holder of Figure(s) [1] to publish the content specifically under the CC BY 4.0 license. 

4. In your Methods section, please provide additional information regarding the permits you obtained for the work. Please ensure you have included the full name of the authority that approved the field site access and, if no permits were required, a brief statement explaining why.

5. We suggest you thoroughly copyedit your manuscript for language usage, spelling, and grammar. If you do not know anyone who can help you do this, you may wish to consider employing a professional scientific editing service.  

Reviewers' comments:

Reviewer's Responses to Questions

**Comments to the Author**

1. Is the manuscript technically sound, and do the data support the conclusions?

Reviewer #1: Partly

Reviewer #2: Partly

2. Has the statistical analysis been performed appropriately and rigorously? 

Reviewer #1: No

Reviewer #2: Yes

3. Have the authors made all data underlying the findings in their manuscript fully available?

Reviewer #1: Yes

Reviewer #2: No

4. Is the manuscript presented in an intelligible fashion and written in standard English?

Reviewer #1: Yes

Reviewer #2: No

5. Review Comments to the Author

Reviewer #1: Rice-Wheat cropping system is a wide spread cultivation system and often considered as one of the oldest known to mankind and we appreciate the author’s effort to execute a revised recommendation of Potassium for Rice-Wheat cropping system under a depleting soil nutrient scenario of different agro ecosystems of Nepal. As this manuscript contains a range of information of different aspects, we feel some re-arrangements and revisions may be beneficial for better comprehension of the content. These may be as follows-

A. In abstract of the manuscript there are mentions of “1. 45 to 75 kg K2O ha-1 in inner-Terai and Terai and 45 to 60 kg ha-1 in high-Hills conditions produced significantly higher grain yields compared to the RDKF.

2. A maximum rate of K fertilizer should not exceed 68 kg K2O ha-1 for rice in all agro-ecozones, 73 kg K2O ha-1 for wheat in inner-Terai and 60 kg K2O ha-1 for wheat in high-Hills and Terai as an economically optimum rate of K fertilizer for crop production.”

Here these two sentences seems to contradict each other in terms of amount of doses in different areas as the first sentence describes K doses for a generalized term “grain yields”, without any specific mention of rice or wheat. Whereas, highest limit mentions of crop specific K doses in the second sentence doesn’t match with the first one.

We will be grateful to receive a mention of Benefit: Cost ratio (BCR) (alongside values for physical and economic optimality) in the Result and Discussion that can potentially explain the economical optimality of the said doses in the Abstract, along with necessary corrections and/or explanations. Please also mention notations for ‘y’ and ‘x’ in the equations of Table 5.

B. In first part of Introduction Winter season of Wheat cultivation has been mentioned as “cold” and “humid”, whereas in Materials and Methods, Winter season has been described as “…generally dry with scanty rainfall.” This also seems to be contradictory and needs necessary corrections.

C. We will be grateful to have K% in FYM used by farmers in Nepal and bibliographic reference of occurrence of K deficiency in field crops in Nepal; as mentioned in the Introduction.

D. We would like to know if there is any application of organic matter in this trial (as application of OM increases nonexchangeable K availability in soil (Pannu et. al., 2003) and would appreciate if Organic Matter (OM) percentage can be expressed as Oxidisable Organic Carbon (OC) percentage as the conversion factor (also known as Van Bemmelen Factor and often taken as 1.72) from OC to OM has been reported to have varied depending upon soil type (Jain et al., 1997) and soil depth (Westman et al., 2006).

E. We will be grateful if instead of detailed writing; values and parameters mentioned under Soil subheading of Materials and Method; could have been arranged under a table and necessary brief description after the table for better comprehension along with bibliographic references of Soil Quality Indices (SQI) mentioned therein. For example –

Initial Soil Parameters(XXX Site) Safe limit/range Current Result Remarks Reference/s for SQI

pH

OC%

Initial Soil Parameters(YYY Site) Safe limit/range Current Result Remarks Reference/s for SQI

pH

OC%

…etc.

Also we will be grateful for explanation of why Total fraction of soil N is estimated whereas only available fractions of P and K were estimated in the Materials and Method.

F. It is interesting to notice that the trials have been executed in three different year schedules for the three different zones. We would be grateful for the reason why the duration is 3years for Rampur site and 2years for the other two sites, as many factors that are evident to affect nutrient availability may vary with time. Please also mention the duration of experiment for each site along with cropping cycle for each year in the Materials and Methods.

G. We would be curious to know the reason behind similar Nitrogen (100 kg.ha-1) and Phosphorous (30 kg.ha-1) doses and split does techniques for both rice and wheat as each of them have different requirements that also varies with soil type and crop cultivation strategies. Please also mention the cultivation practice (like wetland paddy. arable rice cultivation/Terrace Farming/ SRI technique etc…) followed for rice in each region as Potassium movement has been observed to have affected according to periodicity of water regimes in rice-wheat system (Kadrekar and Kibe, 1973).

H. It would be appreciable if the references for protocols that have been used in analyzing soil parameters could have been mentioned in the Materials and Methodology. Please also mention the method of fertilizer application as Potassium movement in soil is reported to have affected by method of application (Nolan and Pritchett, 1960).

I. We will be grateful if you kindly flag (with * or ** at superscript) the correlation coefficients in the text and in supplementary tables according to their significance and mention the version of M.S. Excel and R-software along with R-studio.

J. It would be appreciable if the term “significantly different” can be replaced with “significantly increased/ decreased”, wherever applicable in the text to be more specific. Also, it will be a bit easier to comprehend if before conclusion, the discussion section can be subdivided into three parts, discussing reasons behind the results for each of the areas separately and lastly another part dedicated to discussion for main and important aspects that has come out of the discussion of those three areas.

K. Soil pH is often considered as one of the important factors in determining nutrient availability in soil. Martin et al. (1946) showed at pH values up to 2.5 there was no K fixation, may be due to increased abundance of H3O+ that replace K from the exchange site (Rich and Black, 1964). However, between pH 2.5 and 5.5, the amount of K fixation increased very rapidly. Above pH 5.5, fixation increased more slowly (Martin et al., 1946). As most of the soils of all three areas come under this study come within this pH 5.5 (Table S1), post application of potassium fertilizers, we will be grateful if some discussions can be added on how soil pH has affected the K availability in this study with specifications of sites.

L. In this context, soil pH of the Kabre, under Hilly area is quite low for rice cultivation than soil pH 5.0-6.5 for fertile wetland rice soil as mentioned by Dr. F. N. Ponnamperuma, the celebrated Rice Soil Chemist and former Principal Soil Chemist, International Rice Research Institute (IRRI), Los Banos, Philippines (Ponnamperuma, 1981). As most of the soil functions and microbial nutrient transformations get restricted below 5.5 (Pietri and Brooks, 2008), we will be grateful if this can be emphasized while discussing results of this area.

[ Dr. F. N. Ponnamperuma, the celebrated Rice Soil Chemist and former Principal Soil Chemist, International Rice Research Institute (IRRI), Los Banos, Philippines listed the following following chemical characteristics of fertile wetland rice soils:

pH 5.0-6.5, ECe (mS/cm) <2, Eh (after submergence, in volt) +0.2 to -0.2, org matter (%) 2.0-3.5, total N (%) >0.2, total P (%) >0.02, Olsen P (mg/kg) >10, exch K (mmol/kg) >2, avail S (mg/kg) >10, CEC (mmol/kg) >200, clay composition >50% montmorillonite, active Fe (%) >0.5%, active Mn (%) >0.05, avail Zn (mg/kg) >1, avail B (mg/kg) <5. ]

M. Soil Nitrogen and Phosphorus are also known to affect Potassium availability and uptake by plants. Application of nitrogen and phosphorus are reported to have resulted in 145% increase in potassium uptake as compared to control (Tandon and Sekhon, 1988). Tiwari et al. (1992) reported that response to potassium application in rice increased with increasing rate of nitrogen application. Under nitrogenous fertilization, interlayer potassium (K) can also be replaced by NH4+ as both of them have nearly similar ionic size, influencing its availability. Therefore, it will be appreciable if changes in these factors can be considered while discussing K availability with respect to control in this study.

N. We would appreciate if relations of Plant parameters (yield influencing parameters with yield) with changes in Potassium dosage applied in soil can be discussed in Discussion segment (like how plant height and yield can have a strong positive correlation in terms of changes in potassium content in soil; as revealed in the Result section.)

O. In the discussion section there is a mention “Comparing the K fertilizer consumption under RWCS in the IGP regions, Nepal uses less 324 amount of potassium fertilizer in rice (0.8 kg ha-1) and wheat (1.7 kg ha-1) compared to China in rice (33.2 kg ha-1) and wheat (26.6 kg ha-1).” It is quite unclear here whether the comparison is stated with China, or with IGP. We will be grateful if this could be explained properly. Also mention the region of IGP in bracket wherever the term IGP is mentioned, as IGP is a widespread region stretched amongst countries in Indian subcontinent.

P. We will be grateful if you can justify the term “Revised” in the title of the manuscript; in terms of how this revision is applicable to the whole country of Nepal as these trials are mentioned to have conducted in localized research stations and seem not to have checked in selected progressive farmer’s field with farmer’s packages of practices throughout the country. Also the selected trial sites mainly represent the Central Development Region of Nepal ("Memorial Step of King Mahendra in 1st Poush 2017 BS". reviewnepal.com. 13 December 2017. Retrieved 6 February 2018.), consisting of Province No. 2 (9,661 km²) and Bagmati Pradesh province (20,300 km²) which together share only 20.30% approx. of total land area (147,557 km²) of Nepal. This may have lack of implacability of these doses to other areas of Nepal (distant from these research stations), which may vary in terms of soil-water and land use characteristics. .

Q. It is a pleasure to observe that in the Manuscript; only 11% similarity has been found on Plagiarism Checker X 2018 Professional Edition v6.0.6. (Report Attached). We will be grateful if this can be improved even more

References:

J. C. A. Pietri and P. C. Brookes, “Relationships between soil pH and microbial properties in a UK arable soil,” Soil Biology & Biochemistry, vol. 40, no. 7, pp. 1856–1861, 2008.

Jain, T. B., Graham, R. T. and Adams, D. L. 1997. Carbon to organic matter ratios for soils in Rocky Mountain coniferous forests. Soil Sci. Am. J. 61: 11901195.

Kadrekar, S.B. and Kibe, M. M. (1973). Release of soil potassium on wetting and drying. J. Indian Soc. Soil Sci., 20:231-240.

Martin, J.C., Overstreet, R. and Hoagland, D.R. 1946. Potassium fixation in soils in replaceable and nonreplaceable forms in relation to chemical reactions in the soil. Soil Science Society of America Proceedings 10: 94-101.

Nolan, C.N. and Pritchett, W.L. 1960. Certain factors affecting the leaching of potassium from sandy soils. Proceedings of Soil Crop Science Society of Florida 20: 139-145.

Pannu, R.P.S., Singh, Y., Singh, B. (2003). Effect of long term application of organic materials and inorganic N-fertilizer on potassium fixation and release characteristics of soils under rice-wheat cropping system. J. Potassium Res. (in press)

Ponnamperuma, F.N. (1981). Properties of Tropical Rice Soils. Text of a Series of Lectures Delivery to Graduate Students at the Topical Agriculture College, H. Cardenas, Tabasco, Mexico on 23-25 July 1981.

Rich, C.I., and Black, W.R. 1964. Potassium exchange as affected by cation size, pH, and mineral structure. Soil Science 97: 384-390.

Tandon, H.L.S. and Sekhon, G.S. 1988. Potassium Research and Agricultural Production in India. 144 pp. Fertiliser Development and Consultation Organization, New Delhi.

Tiwari, K.N., Dwivedi, B.S. and Subbarao, A. 1992. Potassium management in rice-wheat system, pp. 93-114. In Pandey, R.K., Dwivedi, B.S. and Sharma, A.K. (eds.) Rice-Wheat Cropping System: Proceedings of Rice-Wheat Workshop, Project Directorate for Cropping Systems Research. Modipuram, India.

Westman, C. J., Hyto¨ nen, J. and Wall, A. 2006. Loss-onignition in the detremination of pools of organic carbon in soils of forests and afforested arable fields. Commun. Soil Sci. Plant Anal. 37: 10591075.

Reviewer #2: The authors Ojha et al. have applied different rates of K fertilizer to estimate crop response to the added K in three agro-ecozones in Nepal where soils are K limited. They used the K rates as K2O, which form of K compound is absent in soil, in reality. The findings do not provide with ground breaking information in K chemistry in soils nor provide with any significant knowledge gaps in K fertilization in crops. It is quite known that in a K limited soil, added K will increase crop production. The statistical method for estimating the optimum K rate is not as robust. As an international and high ranked journal, authors need to provide K uptake, balance and surplus data in your study sites. However, the information bears some merits within the scope of publication in PLOSE ONE. Yet, there are some major flaws in all sections which need to be addressed before it is accepted for publication in PLOSE ONE. My specific comments are added below:

Abstract

Please convert K2O to K and use consistently throughout the paper

Define RDKF L30

What is your hypothesis to use a wide range of K? Did the K content varies that wide?

Mention the duration of work in the abstract L35

L 39, you said maximum should not exceed, rather you should mention what would be the optimum rate for higher economic production

Four times higher than the current rate looks unusually high

Introduction

The aim was yield and nutrient response but nutrient is not only the K, Have you measured K concentrations in grain and straw

Methods

In this section authors should add some initial soil characteristics including K minerals in a table.

L108; define masl

Aadd appropriate references for the weather data given, if it is measured in your station, mention details of the methods of data recording

L137, define OM in its first use

It is better to provide soil quality parameters in a table

Explain the soil sampling time and strategies more details.

Results

Why did you record 1000 grain weight, plant height and panicle length, you have to show the significance of recording these data on the grain and straw yields.

As an international and high ranked journal, you need to provide K uptake, balance and surplus data in your study sites

L187; You evaluated soil properties in different years in different sites which definitely had different climatic conditions, especially rainfall.

L199; where are the graphical presentations of the trend?

You showed yield data from 2009 to up until 2014, but not uniform in different sites?

Table 5; these analysis are based on one year or three years data?

L287; it is not clear in your paper about how did you estimate the residual fertility and its effect on grain yield.

Discussion

L302; this is not clear about what should be the exact K rate

L305, you said about nutrient but should specifically say about K, because you are not estimating other nutrients

L309; witnessed is not appropriate wording here

319; 15 kg what?

L324; potassium, please use K

L327; please update the rates used in Bangladesh etc. it looks much lower than the current rates

overall the discussion doesn’t link very well with your findings.

6. PLOS authors have the option to publish the peer review history of their article (what does this mean?). If published, this will include your full peer review and any attached files.

Reviewer #1: **Yes: **PRAVAT UTPAL ACHARJEE

Reviewer #2: No

---

## [Author Response · Author response to Decision Letter 0]

6 Sep 2020

We also have attached this response letter individually previous in the attachment section.

Journal Requirements Response

[Response]

Thank you so much for reminding us. We have edited as per the guideline.

2. We understand that you purchased seeds and fertilizers from local markets for this study. In your Methods section, please provide additional regarding the source of this material. Please provide the geographic coordinates and names of the purchase locations (e.g., stores, markets), if available, as well as any further details about the purchased items (e.g., lot number, source origin, description of appearance) to ensure reproducibility of the analyses.

[Response]

Details of purchasing store is added in revised manuscript. We contacted the store and asked details, they responded all the inputs were sourced from salt trading corporation. So, we changed it accordingly. We are sorry that we could not be able to access the lot number of the K-fertilizer, however, other details of fertilizers are provided. 

3. We note that [Figure(s) 1] in your submission contain [map/satellite] images which may be copyrighted. All PLOS content is published under the Creative Commons Attribution License (CC BY 4.0), which means that the manuscript, images, and Supporting Information files will be freely available online, and any third party is permitted to access, download, copy, distribute, and use these materials in any way, even commercially, with proper attribution. For these reasons, we cannot publish previously copyrighted maps or satellite images created using proprietary data, such as Google software (Google Maps, Street View, and Earth). For more information, see our copyright guidelines: http://journals.plos.org/plosone/s/licenses-and-copyright.

 1. You may seek permission from the original copyright holder of Figure(s) [1] to publish the content specifically under the CC BY 4.0 license. 

[Response]

Figure 1 (according to older version of manuscript) was removed as suggested.

4. In your Methods section, please provide additional information regarding the permits you obtained for the work. Please ensure you have included the full name of the authority that approved the field site access and, if no permits were required, a brief statement explaining why.

[Response]

This was a regular project of Nepal Agricultural Research Council and a year round proposal was called which is reviewed internally and by reviewing committee. If the committee pass the proposal NARC headquarter will allocate the budget. We have updated this information in methodology section. 

5. We suggest you thoroughly copyedit your manuscript for language usage, spelling, and grammar. If you do not know anyone who can help you do this, you may wish to consider employing a professional scientific editing service. 

[Response]

English editing has been carried out by native American professional. His name is included in Acknowledgement section. In addition, manuscript in track change mode shows his English editing work besides our revision to respond comments made by reviewers. 

Response to Reviewer 1

Rice-Wheat cropping system is a wide spread cultivation system and often considered as one of the oldest known to mankind and we appreciate the author’s effort to execute a revised recommendation of Potassium for Rice-Wheat cropping system under a depleting soil nutrient scenario of different agro ecosystems of Nepal. As this manuscript contains a range of information of different aspects, we feel some re-arrangements and revisions may be beneficial for better comprehension of the content. These may be as follows-

[Response]

Dear reviewer, 

We would like to thank you very much for your valuable time for reviewing this manuscript. Yes, we agree, your insightful suggestion has improved our manuscript’s quality.

A. In abstract of the manuscript there are mentions of 

1. 45 to 75 kg K2O ha-1 in inner-Terai and Terai and 45 to 60 kg ha-1 in high-Hills conditions produced significantly higher grain yields compared to the RDKF. 

2. A maximum rate of K fertilizer should not exceed 68 kg K2O ha-1 for rice in all agro-ecozones, 73 kg K2O ha-1 for wheat in inner-Terai and 60 kg K2O ha-1 for wheat in high-Hills and Terai as an economically optimum rate of K fertilizer for crop production.”

Here these two sentences seems to contradict each other in terms of amount of doses in different areas as the first sentence describes K doses for a generalized term “grain yields”, without any specific mention of rice or wheat. Whereas, highest limit mentions of crop specific K doses in the second sentence doesn’t match with the first one. 

[Response]

Yes, we agreed. Here we are trying to show two different results. First, a mean response of the crop and second an economic optimum dose of fertilizer. We have improved it in the revised manuscript to avoid confusion. Thank you again for dragging our attention.

We will be grateful to receive a mention of Benefit: Cost ratio (BCR) (alongside values for physical and economic optimality) in the Result and Discussion that can potentially explain the economical optimality of the said doses in the Abstract, along with necessary corrections and/or explanations. Please also mention notations for ‘y’ and ‘x’ in the equations of Table 5.

[Response]

Following your suggestion, we considered unit price of input and output to calculate economic optimality of the fertilizer dose, also required for BCR. The concept of calculating economic optimum fertilizer dose by derivation the output input ratio with price ratio (equation 3) provides a stronger recommendation base than BCR which is also revisable in temporal scale. So, we prefer to go with this derivative approach. 

Notation for X and Y in Table 5 equation corrected.

B. In first part of Introduction Winter season of Wheat cultivation has been mentioned as “cold” and “humid”, whereas in Materials and Methods, Winter season has been described as “…generally dry with scanty rainfall.” This also seems to be contradictory and needs necessary corrections.

[Response]

Corrected in the manuscript. Thank you for your suggestion. 

C. We will be grateful to have K% in FYM used by farmers in Nepal and bibliographic reference of occurrence of K deficiency in field crops in Nepal; as mentioned in the Introduction. 

[Response]

N,P,K contents of FYM were added in the manuscript. 

D. We would like to know if there is any application of organic matter in this trial (as application of OM increases nonexchangeable K availability in soil (Pannu et. al., 2003) and would appreciate if Organic Matter (OM) percentage can be expressed as Oxidisable Organic Carbon (OC) percentage as the conversion factor (also known as Van Bemmelen Factor and often taken as 1.72) from OC to OM has been reported to have varied depending upon soil type (Jain et al., 1997) and soil depth (Westman et al., 2006).

[Response]

No organic fertilizer (manure) were added in the research plots. Our national recommendation is based on OM content. So, we prefer to present our result in OM and maintained consistency throughout the text. To address your concern, we convert the OM to OC with conversion factor in preliminary result table (Table 1) and continued to use OM value in the rest of the manuscript. 

E. We will be grateful if instead of detailed writing; values and parameters mentioned under Soil subheading of Materials and Method; could have been arranged under a table and necessary brief description after the table for better comprehension along with bibliographic references of Soil Quality Indices (SQI) mentioned therein. For example –

Initial Soil Parameters(XXX Site) Safe limit/range Current Result Remarks Reference/s for SQI

pH 

OC% 

Initial Soil Parameters(YYY Site) Safe limit/range Current Result Remarks Reference/s for SQI

pH 

OC% 

…etc.

Also we will be grateful for explanation of why Total fraction of soil N is estimated whereas only available fractions of P and K were estimated in the Materials and Method.

[Response]

Table has been updated accordingly. Our central soil laboratory made the recommendation on the basis of total N and available P and K basis. So, we adopt the same recommendation for the uniformity.

F. It is interesting to notice that the trials have been executed in three different year schedules for the three different zones. We would be grateful for the reason why the duration is 3years for Rampur site and 2years for the other two sites, as many factors that are evident to affect nutrient availability may vary with time. Please also mention the duration of experiment for each site along with cropping cycle for each year in the Materials and Methods.

[Response]

Our objective was to see the nutrient response only in two crop cycle. 

G. We would be curious to know the reason behind similar Nitrogen (100 kg.ha-1) and Phosphorous (30 kg.ha-1) doses and split does techniques for both rice and wheat as each of them have different requirements that also varies with soil type and crop cultivation strategies. Please also mention the cultivation practice (like wetland paddy. arable rice cultivation/Terrace Farming/ SRI technique etc…) followed for rice in each region as Potassium movement has been observed to have affected according to periodicity of water regimes in rice-wheat system (Kadrekar and Kibe, 1973).

[Response]

Yes we agree with your opinion. 21 days old seedling of rice was transplanted in the puddled field (water submerged field) and wheat seeds were directly sown in the field. We have mentioned this cultivation practices in our manuscript.

H. It would be appreciable if the references for protocols that have been used in analyzing soil parameters could have been mentioned in the Materials and Methodology. Please also mention the method of fertilizer application as Potassium movement in soil is reported to have affected by method of application (Nolan and Pritchett, 1960).

[Response]

Yes, we agreed, and it has been corrected in revised manuscript. 

I. We will be grateful if you kindly flag (with * or ** at superscript) the correlation coefficients in the text and in supplementary tables according to their significance and mention the version of M.S. Excel and R-software along with R-studio.

[Response]

This has been corrected in the revised manuscript and supplementary table.

J. It would be appreciable if the term “significantly different” can be replaced with “significantly increased/ decreased”, wherever applicable in the text to be more specific. Also, it will be a bit easier to comprehend if before conclusion, the discussion section can be subdivided into three parts, discussing reasons behind the results for each of the areas separately and lastly another part dedicated to discussion for main and important aspects that has come out of the discussion of those three areas. 

[Response]

This has been corrected in the revised manuscript. 

K. Soil pH is often considered as one of the important factors in determining nutrient availability in soil. Martin et al. (1946) showed at pH values up to 2.5 there was no K fixation, may be due to increased abundance of H3O+ that replace K from the exchange site (Rich and Black, 1964). However, between pH 2.5 and 5.5, the amount of K fixation increased very rapidly. Above pH 5.5, fixation increased more slowly (Martin et al., 1946). As most of the soils of all three areas come under this study come within this pH 5.5 (Table S1), post application of potassium fertilizers, we will be grateful if some discussions can be added on how soil pH has affected the K availability in this study with specifications of sites.

[Response]

Wording issue has been corrected in revised manuscript. We have added the discussion part as suggested. Thank you very much.

L. In this context, soil pH of the Kabre, under Hilly area is quite low for rice cultivation than soil pH 5.0-6.5 for fertile wetland rice soil as mentioned by Dr. F. N. Ponnamperuma, the celebrated Rice Soil Chemist and former Principal Soil Chemist, International Rice Research Institute (IRRI), Los Banos, Philippines (Ponnamperuma, 1981). As most of the soil functions and microbial nutrient transformations get restricted below 5.5 (Pietri and Brooks, 2008), we will be grateful if this can be emphasized while discussing results of this area.

[ Dr. F. N. Ponnamperuma, the celebrated Rice Soil Chemist and former Principal Soil Chemist, International Rice Research Institute (IRRI), Los Banos, Philippines listed the following following chemical characteristics of fertile wetland rice soils:

pH 5.0-6.5, ECe (mS/cm) <2, Eh (after submergence, in volt) +0.2 to -0.2, org matter (%) 2.0-3.5, total N (%) >0.2, total P (%) >0.02, Olsen P (mg/kg) >10, exch K (mmol/kg) >2, avail S (mg/kg) >10, CEC (mmol/kg) >200, clay composition >50% montmorillonite, active Fe (%) >0.5%, active Mn (%) >0.05, avail Zn (mg/kg) >1, avail B (mg/kg) <5. ]

[Response]

Thank you very much for suggestion. These have been included in Discussion section of revised manuscript.

M. Soil Nitrogen and Phosphorus are also known to affect Potassium availability and uptake by plants. Application of nitrogen and phosphorus are reported to have resulted in 145% increase in potassium uptake as compared to control (Tandon and Sekhon, 1988). Tiwari et al. (1992) reported that response to potassium application in rice increased with increasing rate of nitrogen application. Under nitrogenous fertilization, interlayer potassium (K) can also be replaced by NH4+ as both of them have nearly similar ionic size, influencing its availability. Therefore, it will be appreciable if changes in these factors can be considered while discussing K availability with respect to control in this study.

[Response]

Suggestions have been included in the discussion section.

N. We would appreciate if relations of Plant parameters (yield influencing parameters with yield) with changes in Potassium dosage applied in soil can be discussed in Discussion segment (like how plant height and yield can have a strong positive correlation in terms of changes in potassium content in soil; as revealed in the Result section.) 

[Response]

Suggestions have been included in the discussion section.

O. In the discussion section there is a mention “Comparing the K fertilizer consumption under RWCS in the IGP regions, Nepal uses less 324 amount of potassium fertilizer in rice (0.8 kg ha-1) and wheat (1.7 kg ha-1) compared to China in rice (33.2 kg ha-1) and wheat (26.6 kg ha-1).” It is quite unclear here whether the comparison is stated with China, or with IGP. We will be grateful if this could be explained properly. Also mention the region of IGP in bracket wherever the term IGP is mentioned, as IGP is a widespread region stretched amongst countries in Indian subcontinent. 

[Response]

Corrected in the revised manuscript accordingly. 

P. We will be grateful if you can justify the term “Revised” in the title of the manuscript; in terms of how this revision is applicable to the whole country of Nepal as these trials are mentioned to have conducted in localized research stations and seem not to have checked in selected progressive farmer’s field with farmer’s packages of practices throughout the country. Also the selected trial sites mainly represent the Central Development Region of Nepal ("Memorial Step of King Mahendra in 1st Poush 2017 BS". reviewnepal.com. 13 December 2017. Retrieved 6 February 2018.), consisting of Province No. 2 (9,661 km²) and Bagmati Pradesh province (20,300 km²) which together share only 20.30% approx. of total land area (147,557 km²) of Nepal. This may have lack of implacability of these doses to other areas of Nepal (distant from these research stations), which may vary in terms of soil-water and land use characteristics. .

[Response]

Very good observation. We fully agree with your point and are careful about it. We are trying to cover all the ecological belt, however, it is restricted to a single province only. The soil we have studied is Inceptisol (a part is mentioned in Introduction) which is dominant in most of the ecological belts of Nepal. Henceforth, we attempt to say it as a ‘revised’ K rate in Nepal. We considered your suggestion and revised our recommendation stating ‘limitation of this study’ to different soil types in different agro-ecological belts. 

Q. It is a pleasure to observe that in the Manuscript; only 11% similarity has been found on Plagiarism Checker X 2018 Professional Edition v6.0.6. (Report Attached). We will be grateful if this can be improved even more

[Response]

While reviewing, we tried to fix it at our best. Thank you very much for this intensive and caring review.

Response to Reviewer 2

The authors Ojha et al. have applied different rates of K fertilizer to estimate crop response to the added K in three agro-ecozones in Nepal where soils are K limited. They used the K rates as K2O, which form of K compound is absent in soil, in reality. The findings do not provide with ground breaking information in K chemistry in soils nor provide with any significant knowledge gaps in K fertilization in crops. It is quite known that in a K limited soil, added K will increase crop production. The statistical method for estimating the optimum K rate is not as robust. As an international and high ranked journal, authors need to provide K uptake, balance and surplus data in your study sites. However, the information bears some merits within the scope of publication in PLOSE ONE. Yet, there are some major flaws in all sections which need to be addressed before it is accepted for publication in PLOSE ONE. My specific comments are added below:

[Response]

Dear reviewer, 

We are highly thankful for your time to review our manuscript. Your insightful suggestion/ comments have improved our manuscript’s quality.

Abstract

Please convert K2O to K and use consistently throughout the paper

[Response]

Thank for recommending this conversion. In our national context, all the fertilizer recommendations were made on K2O basis and soil test results also presented in K2O basis. So, we followed this pattern. To address your concern, we provide the conversion factor of K to K2O in Table 1. 

Define RDKF L30

[Response]

Corrected in the revised manuscript accordingly. 

What is your hypothesis to use a wide range of K? Did the K content varies that wide?

[Response]

The hypothesis was mainly based on the field observation as we found rice-wheat cropping system is mostly affected by low level of K. We have mentioned this in Introduction section.

Mention the duration of work in the abstract L35

[Response]

Corrected in the revised manuscript accordingly. 

L 39, you said maximum should not exceed, rather you should mention what would be the optimum rate for higher economic production

Four times higher than the current rate looks unusually high

[Response]

Corrected in the revised manuscript accordingly. 

Introduction

The aim was yield and nutrient response but nutrient is not only the K, Have you measured K concentrations in grain and straw

[Response]

We measured the K uptake in grain and straw of a single location (inner-Terai) only. This was also not significant in any parameters. So, decided not to present in the result. 

Methods

In this section authors should add some initial soil characteristics including K minerals in a table.

[Response]

Inserted as Table 1.

L108; define masl

[Response]

Defined in the revised manuscript accordingly. 

Add appropriate references for the weather data given, if it is measured in your station, mention details of the methods of data recording

[Response]

Corrected in the revised manuscript accordingly. 

L137, define OM in its first use

[Response]

Corrected in the revised manuscript accordingly. 

It is better to provide soil quality parameters in a table

[Response]

Inserted in Table 1 of the revised manuscript. 

Explain the soil sampling time and strategies more details.

[Response]

Explained in the text.

Results

Why did you record 1000 grain weight, plant height and panicle length, you have to show the significance of recording these data on the grain and straw yields.

[Response]

These are our attributed parameters. We have showed the correlation between these parameters with the yields in supplementary files. 

As an international and high ranked journal, you need to provide K uptake, balance and surplus data in your study sites 

[Response]

We measured the K uptake in grain and straw of a single location (inner-Terai) only. This was also not significant in any parameters. So, we decided not to present in the result. 

L187; You evaluated soil properties in different years in different sites which definitely had different climatic conditions, especially rainfall.

L199; where are the graphical presentations of the trend?

[Response]

We presented data averaged over the year. Our trial was conducted 1-2 years in each location where we found similar trend data in each months over the years. To reduce the number of image we have presented it as a text. 

You showed yield data from 2009 to up until 2014, but not uniform in different sites?

[Response]

Yes, we had started our work from 2009 but due to logistic management we had conducted our trial in different years in three locations. So, it ended up in 2014. We have described the timeline in the methodology.

Table 5; these analysis are based on one year or three years data? 

[Response]

The analysis based on the average data over the years. Inserted an explanation as a note below the table. 

L287; it is not clear in your paper about how did you estimate the residual fertility and its effect on grain yield.

[Response]

It is the yield obtained from the control plot. Yes, we agreed that it is inherent fertility, word changed and explained the term in the parentheses. 

Discussion

L302; this is not clear about what should be the exact K rate

[Response]

We are trying to say K fertilizer rate range 50 – 60 Kg K2O per ha. Corrected.

L305, you said about nutrient but should specifically say about K, because you are not estimating other nutrients

[Response]

Corrected in the revised manuscript accordingly. 

L309; witnessed is not appropriate wording here

[Response]

Corrected in the revised manuscript accordingly. 

319; 15 kg what?

[Response]

Corrected in the revised manuscript accordingly. 

L324; potassium, please use K

[Response]

Corrected in the revised manuscript accordingly. 

L327; please update the rates used in Bangladesh etc. it looks much lower than the current rates

[Response]

In terms of K rate (not as K2O) seems OK to us.

Overall the discussion doesn’t link very well with your findings.

[Response]

We tried to add and edit our best as suggested by next reviewer. Hope it looks better than previous version.

---

## [Decision Letter · Decision Letter 1]

23 Nov 2020

PONE-D-20-12836R1

Revised potassium nutrition in rice-wheat cropping system of Nepal

PLOS ONE

Dear Dr. Ojha,

Thank you for submitting your manuscript to PLOS ONE. After careful consideration, we feel that it has merit but does not fully meet PLOS ONE’s publication criteria as it currently stands. Therefore, we invite you to submit a revised version of the manuscript that addresses the points raised during the review process.

We look forward to receiving your revised manuscript.

Kind regards,

Vassilis G. Aschonitis

Academic Editor

PLOS ONE

Reviewers' comments:

Reviewer's Responses to Questions

**Comments to the Author**

1. If the authors have adequately addressed your comments raised in a previous round of review and you feel that this manuscript is now acceptable for publication, you may indicate that here to bypass the “Comments to the Author” section, enter your conflict of interest statement in the “Confidential to Editor” section, and submit your "Accept" recommendation.

Reviewer #1: (No Response)

Reviewer #2: All comments have been addressed

2. Is the manuscript technically sound, and do the data support the conclusions?

Reviewer #1: Partly

Reviewer #2: Yes

3. Has the statistical analysis been performed appropriately and rigorously? 

Reviewer #1: Yes

Reviewer #2: Yes

4. Have the authors made all data underlying the findings in their manuscript fully available?

Reviewer #1: Yes

Reviewer #2: Yes

5. Is the manuscript presented in an intelligible fashion and written in standard English?

Reviewer #1: Yes

Reviewer #2: Yes

6. Review Comments to the Author

Reviewer #1: We are grateful to receive necessary modifications of the manuscript entitled “Revised potassium nutrition in rice-wheat cropping system of Nepal” by Ojha et. el. Although the authors have employed considerable efforts for the improvement, we have observed some points from the previous suggestions have not been addressed properly. We would appreciate if the authors could consider the previous ones as well as the new ones as follows to reconstruct the manuscript.

1. We understand that in the abstract section the authors described a mean response of Potassium fertilization, irrespective of rice/wheat cultivated along with doses for economic optimum. However, a mean fertilizer response value for two different crops, having different potassium requirement (http://www.fao.org/3/a0443e/a0443e04.pdf ), may still create confusion. It would be, therefore, appreciable that the authors avoid the ‘mean response’ part from the abstract as well as from the result/discussion section.

2. We appreciate the calculation of “Economic optimum” and would be grateful to receive a bibliographic reference in support of the calculation of “Economic optimum”.

3. In Materials and Methods, winter season has still been mentioned as “…generally cold and humid with scarce rainfall.” We would appreciate if authors could mention that how “Humidity” can be related with “Scarcity of rainfall”?

4. We are grateful for the sincere effort of the authors, applied for construction of Table-1. However, we feel that there may be lack of comprehensibility in the previous suggestion, that the Van Bemmelen Factor (often taken as 1.72) varies with soil type (Jain et al., 1997) and soil depth (Westman et al., 2006), leading to varying results of OM depending upon soil type and depth. It is, therefore highly appreciable to express the OM content as OC throughout the manuscript. Regarding Total soil Nitrogen analysis, we will be grateful to have bibliographic mention of the recommendation of the “Central Soil Laboratory” procedures of soil analysis. Please also provide bibliographic references to the analytical protocols (like Olsen, Walkley-Black etc…) mentioned.

5. In table number 5, one of the data series under column “straw yield” is mentioned to be recorded in 2012. Please rectify the ‘year’ if this is a flaw or else explain.

6. Regarding suggestion tagged under “G”, we appreciate that the cultivation practices of rice and wheat is mentioned in the text. However, we are still curious to know why similar dose and split dose techniques for Nitrogen (100 kg.ha-1) and Phosphorous (30 kg.ha-1) were applied for both rice and wheat as each one of them have different nutrient requirements (http://www.fao.org/3/a0443e/a0443e04.pdf ). Please address this query.

7. Please mention table number and/or figure number along with respective findings (value of economic optimum, K doses and yield parameters) in the Discussion section.

8. It is quite appreciable that the authors modified the discussion part. However, a major portion of the discussion still engages in establishing “Why” the additional dose of Potassium may be useful, whereas, it misses “How” it may be useful, to a great extent. Please address point “N” from the previous suggestion more clearly and relate the findings with applied potassium dosages (like why there is a 30-33% rice yield decline in Rampur site in 2nd year? Why there is a 38% grain yield and 20% straw yield decline over 3 years in Rampur site? How applied potassium influenced all these increase and/or decrease in all these three experimental sites? What role potassium is playing in soil and/or plant body to increase and/or decrease yield in all these three sites specifically? Please relate these kinds of major findings with applied potassium and include in the discussion portion).

9. Please reconstruct the conclusion part mentioning Potassium recommendation according to economic optimum for both Rice and Wheat for all three experimental regions. It is highly advisable to mention those recommendations according to the crops and varieties put under experimental trials instead of mentioning to be fruitful for any other hybrid varieties not tested in this experiment. Please also rectify this in the Discussion part as well.

10. Regarding point “F” of our previous queries, we will be grateful to receive a more clear response on why the duration is 3 years for Rampur site and One year less (2 years) for the other two sites. Please mention crop cycles in detail (each year; each site), crops cultivated in those fields and fertilizers previously used.

11. We understand and appreciate the way the authors mentioned limitations of this study in the text. However, we feel that the mentions of limitations of research itself in the text may question the relevance of the research hypothesis. Please express the limitations as Future scope/s of Research and avoid words like “pitfall”, “limitations” etc. And, since it is not very convincing to declare the resulted dose to be revised without being rigorously tested in farmer’s fields (where soil characteristics may differ from the research stations depending upon farmer’s packages of cultivation practices and crop cycles) of the representative agro-eco zones of Nepal, we will be grateful, instead, if the authors could modify the title, by including, what this research is basically aimed at according to the objectives. [ like “Effect of increased dose of Potassium over recommendation on yield and nutrient response of rice and wheat in High hill, Inner-terai and Terai region of Nepal” or whichever is convenient.]

12. We really appreciate the hard work of the authors employed on reducing plagiarism of this text from 11% to 9% as found on Plagiarism Checker X 2018 Professional Edition v6.0.6. (Report attached).

References:

Jain, T. B., Graham, R. T. and Adams, D. L. 1997. Carbon to organic matter ratios for soils in Rocky Mountain coniferousforests. Soil Sci. Am. J. 61: 11901195.

Westman, C. J., Hyto¨ nen, J. and Wall, A. 2006.Loss-onignitionin the detremination of pools of organic carbon insoils of forests and afforested arable fields.Commun. Soil Sci.Plant Anal. 37: 10591075.

Reviewer #2: I am not yet convinced of keeping the K form as K2O, which is not a real K form, it is not available in soil either. As an international journal this form should be uniform as of other journals.

7. PLOS authors have the option to publish the peer review history of their article (what does this mean?). If published, this will include your full peer review and any attached files.

Reviewer #1: **Yes: **PRAVAT UTPAL ACHARJEE

Reviewer #2: No

---

## [Author Response · Author response to Decision Letter 1]

6 Dec 2020

We are grateful to receive necessary modifications of the manuscript entitled “Revised potassium nutrition in rice-wheat cropping system of Nepal” by Ojha et. el. Although the authors have employed considerable efforts for the improvement, we have observed some points from the previous suggestions have not been addressed properly. We would appreciate if the authors could consider the previous ones as well as the new ones as follows to reconstruct the manuscript.

[Response]: We would like to thank you very much for this second round review and your effort to make our work a standard quality. We are trying to address each and every query in details. 

We understand that in the abstract section the authors described a mean response of Potassium fertilization, irrespective of rice/wheat cultivated along with doses for economic optimum. However, a mean fertilizer response value for two different crops, having different potassium requirement (http://www.fao.org/3/a0443e/a0443e04.pdf ), may still create confusion. It would be, therefore, appreciable that the authors avoid the ‘mean response’ part from the abstract as well as from the result/discussion section.

[Response]: We thank you for your valuable comment and suggestion. The pdf file was helpful to clarify our concept. We have removed the ‘mean response’ word from all the parts of the manuscript accordingly. 

1. We appreciate the calculation of “Economic optimum” and would be grateful to receive a bibliographic reference in support of the calculation of “Economic optimum”.

[Response]: We have provided the reference [40] as a reference to calculate the Economic optimum and the reference is provided. 

2. In Materials and Methods, winter season has still been mentioned as “…generally cold and humid with scarce rainfall.” We would appreciate if authors could mention that how “Humidity” can be related with “Scarcity of rainfall”?

[Response]: Yes, we agree with your argument. However, several literatures referred winter as cold and humid. In fact, we received low temp (<10 0C) and high relative humidity (>90-95%). Rainfall is scarce compared to monsoon but is enough to maintain the humidity. So, we think it is clear. To avoid confusion, we have removed ‘scarce rainfall’ from that sentence. 

3. We are grateful for the sincere effort of the authors, applied for construction of Table-1. However, we feel that there may be lack of comprehensibility in the previous suggestion, that the Van Bemmelen Factor (often taken as 1.72) varies with soil type (Jain et al., 1997) and soil depth (Westman et al., 2006), leading to varying results of OM depending upon soil type and depth. It is, therefore highly appreciable to express the OM content as OC throughout the manuscript. Regarding Total soil Nitrogen analysis, we will be grateful to have bibliographic mention of the recommendation of the “Central Soil Laboratory” procedures of soil analysis. Please also provide bibliographic references to the analytical protocols (like Olsen, Walkley-Black etc…) mentioned.

[Response]: We have changed OM to OC throughout the manuscript as per your suggestion. We also have provided the bibliographic references of the analytical protocols. 

4. In table number 5, one of the data series under column “straw yield” is mentioned to be recorded in 2012. Please rectify the ‘year’ if this is a flaw or else explain.

[Response]: We are very grateful to the reviewer to spot this kind of detail review. Thank you very much. We have corrected it. 

5. Regarding suggestion tagged under “G”, we appreciate that the cultivation practices of rice and wheat is mentioned in the text. However, we are still curious to know why similar dose and split dose techniques for Nitrogen (100 kg.ha-1) and Phosphorous (30 kg.ha-1) were applied for both rice and wheat as each one of them have different nutrient requirements (http://www.fao.org/3/a0443e/a0443e04.pdf ). Please address this query.

[Response]: We again thank you for repeating this query about N and P dose and application method similar for rice and wheat. This is the blanket national recommendation for rice and wheat and we have followed the same recommendation. As you see the K rate is also recommended the same for rice and wheat. Our objective was to recommend a revised K rate in place of this blanket K rate. Our finding will be valuable to convince our policy maker to make a separate and robust recommendation in each agro-zone and in each crop type. 

6. Please mention table number and/or figure number along with respective findings (value of economic optimum, K doses and yield parameters) in the Discussion section.

[Response]: We added the table and figure reference in the respective findings of the discussion section. 

7. It is quite appreciable that the authors modified the discussion part. However, a major portion of the discussion still engages in establishing “Why” the additional dose of Potassium may be useful, whereas, it misses “How” it may be useful, to a great extent. Please address point “N” from the previous suggestion more clearly and relate the findings with applied potassium dosages (like why there is a 30-33% rice yield decline in Rampur site in 2nd year? Why there is a 38% grain yield and 20% straw yield decline over 3 years in Rampur site? How applied potassium influenced all these increase and/or decrease in all these three experimental sites? What role potassium is playing in soil and/or plant body to increase and/or decrease yield in all these three sites specifically? Please relate these kinds of major findings with applied potassium and include in the discussion portion).

[Response]: You are right, how additional doses of K fertilizer is used by produced biomass or grain can be included in Discussion section. However, it was our limitation (we did not run grain/ biomass K analysis) as well as out of scope (we focused on application of different K rates on crop yield and soil properties). Besides, in our Results section- 3.1: Soil chemical properties, we have mentioned that no significant changes in soil N concentration due to addition of different K rates. So, we were unable to discussion further about N and K relationship in Discussion section have already mentioned as follow-

L#386-388: The current study reported that there were no differences in soil total N after a year’s addition of K fertilizer, suggesting no adverse effects on total N due to treatment application.”

We added following sentence to make it clear—

L#388-390: “However, our study limits that we did not observe the effects of different rates of K fertilizer on uptake of nutrients from the soil by plant roots for grain yield and biomass production.”

8. Please reconstruct the conclusion part mentioning Potassium recommendation according to economic optimum for both Rice and Wheat for all three experimental regions. It is highly advisable to mention those recommendations according to the crops and varieties put under experimental trials instead of mentioning to be fruitful for any other hybrid varieties not tested in this experiment. Please also rectify this in the Discussion part as well.

[Response]: We have added the potassium recommendation according to our economic optimum value. We also have removed the part of ‘high yielding and improved varieties” part from the conclusion. Yes, we believe it helps to put our recommendation in a stronger way. 

9. Regarding point “F” of our previous queries, we will be grateful to receive a more clear response on why the duration is 3 years for Rampur site and One year less (2 years) for the other two sites. Please mention crop cycles in detail (each year; each site), crops cultivated in those fields and fertilizers previously used. 

[Response]: There is no typical objective to take 3 years measurement from Rampur site and 2 years measurement from other two sites. Rampur site is located in accessible area, so a three year harvest is possible. Other two sites, only 2 years measurement is possible due to lack of accessibility. We are trying to be honest to put all the data from all sites that we have taken during the research period. We understand the concern of reviewer that three year data is more reliable but we hope you will understand and accept our condition. We thought two year data is enough to justify our objective to monitor the crop response with additional K rate. As our objective is not to compare between the locations, this year difference doesn’t affect our statistical analysis. 

10. We understand and appreciate the way the authors mentioned limitations of this study in the text. However, we feel that the mentions of limitations of research itself in the text may question the relevance of the research hypothesis. Please express the limitations as Future scope/s of Research and avoid words like “pitfall”, “limitations” etc. And, since it is not very convincing to declare the resulted dose to be revised without being rigorously tested in farmer’s fields (where soil characteristics may differ from the research stations depending upon farmer’s packages of cultivation practices and crop cycles) of the representative agro-eco zones of Nepal, we will be grateful, instead, if the authors could modify the title, by including, what this research is basically aimed at according to the objectives. [ like “Effect of increased dose of Potassium over recommendation on yield and nutrient response of rice and wheat in High hill, Inner-terai and Terai region of Nepal” or whichever is convenient.] 

[Response]: 

We fully agree with the reviewer, the word ‘pitfall’ reduce the strength of our work. So, we have removed it in the revised manuscript. Now, we realise it provides a great sense. Thank you very much. 

Our objective is to recommend a new rate of K dose specific to the agro-ecozones in rice and wheat cropping system. Yes, definitely rigorous field trial is necessary but we think “Revision of potassium” is right keyword in the topic. Yes, we definitely agree, the word ‘nutrition’ is not suitable as you suggested in the previous comment (8). We are decided to change it as ‘rate’. And of course it is representative of “Terai, inner-terai, and high-hills” which you have suggested to include it as well. We replaced it with word “agro-ecozones” to shorten the title word count. So, our revised title is “Revision of potassium rate in rice-wheat cropping system in three agro-ecozones of Nepal”. 

11. We really appreciate the hard work of the authors employed on reducing plagiarism of this text from 11% to 9% as found on Plagiarism Checker X 2018 Professional Edition v6.0.6. (Report attached).

[Response]: Thank you very much for checking this one of the important publication integrity components. We highly appreciate your sincere review. 

References:

Jain, T. B., Graham, R. T. and Adams, D. L. 1997. Carbon to organic matter ratios for soils in Rocky Mountain coniferousforests. Soil Sci. Am. J. 61: 11901195.

Westman, C. J., Hyto¨ nen, J. and Wall, A. 2006.Loss-onignitionin the detremination of pools of organic carbon insoils of forests and afforested arable fields.Commun. Soil Sci.Plant Anal. 37: 10591075.

---

## [Decision Letter · Decision Letter 2]

4 Jan 2021

PONE-D-20-12836R2

Revision of potassium rate in Rice-Wheat cropping system at three agro-ecozones of Nepal

PLOS ONE

Dear Dr. Ojha,

Thank you for submitting your manuscript to PLOS ONE. After careful consideration, we feel that it has merit but does not fully meet PLOS ONE’s publication criteria as it currently stands. Therefore, we invite you to submit a revised version of the manuscript that addresses the points raised during the review process.

This decidion is based on the fact that we received to completely contrasting decisions, one acceptance and one rejection. We believe that the comments of the reviewer who rejected the paper are managable and that you could provide a revision solving the indicated issues by the reviewer. 

We look forward to receiving your revised manuscript.

Kind regards,

Vassilis G. Aschonitis

Academic Editor

PLOS ONE

Reviewers' comments:

Reviewer's Responses to Questions

**Comments to the Author**

1. If the authors have adequately addressed your comments raised in a previous round of review and you feel that this manuscript is now acceptable for publication, you may indicate that here to bypass the “Comments to the Author” section, enter your conflict of interest statement in the “Confidential to Editor” section, and submit your "Accept" recommendation.

Reviewer #1: (No Response)

Reviewer #2: All comments have been addressed

2. Is the manuscript technically sound, and do the data support the conclusions?

Reviewer #1: Partly

Reviewer #2: Yes

3. Has the statistical analysis been performed appropriately and rigorously? 

Reviewer #1: No

Reviewer #2: Yes

4. Have the authors made all data underlying the findings in their manuscript fully available?

Reviewer #1: No

Reviewer #2: Yes

5. Is the manuscript presented in an intelligible fashion and written in standard English?

Reviewer #1: Yes

Reviewer #2: Yes

6. Review Comments to the Author

Reviewer #1: We appreciate constant and sincere effort of the authors in making the revised manuscript better. However, in case of some responses made by the authors, it is hard for us to agree. These are as follows-

1. Regarding the response of point 10, we would like to mention that we have never suggested the term “nutrition” to be ‘not suitable’ in our comment number 8. Instead we mentioned that we would prefer if the authors include the terms “nutrient response” and “yield response” into the title. We urge the authors to be cautious on making such acquisitions.

We understand the author’s concern that a revised recommendation will help in making accurate policies in the country. However, as mentioned earlier, it is not only unscientific, but also unethical to declare dose/s of potassium as revised for the entire country based on experimental research in only 3 research stations, without considering being crossed checked the results in farmer’s field even for once! Rigorous field test is a must for any recommendation.

Furthermore, Potassium, being an active cation, also influences dynamics of other nutrient ions in soil as well as in plant body. For example, Potassium indirectly influences B availability due to its effect on Ca absorption (Mattigod et al. 1985) and it is well known that a very narrow range of deficiency and toxicity of boron in plant body can severely affect the yield. This is just one example… there could be many more!

As mentioned in response of point 7, the authors have provided no robust explanations in discussions of their own findings (like why there is a 30-33% rice yield decline in Rampur site in 2nd year? Why there is a 38% grain yield and 20% straw yield decline over 3 years in Rampur site? Etc…). And as the researchers did not check for the dynamics and uptake of potassium content in plant parts, there is no direct evidence of how potassium influenced the yield in this experiment. The limitation of this work itself inhibits from calling it a ‘Revision’.

Therefore, we could not be more positive here for the publication of this article due to aforesaid reasons and absence of the author’s effort in modifying these key concerns even after two revision attempts and we would like to rely upon the respected Editors for the final decision.

We really appreciate and would like to thank the authors for the other necessary modifications made so far. Although this manuscript has some merit regarding positive responses of potassium application on rice and wheat; the authors should be more considerate in segregating research hypothesis in context with the unexplained portions from result and discussion segment and need to reconstruct the title according to the objectives achieved through this research.

Some more points that could be taken under consideration in future are as follows-

2. Bibliographic reference of “Economic optimum” as provided by the authors is not robust enough to understand the application of this procedure in the present context. We have downloaded and read the book which includes some implications of “economic optimum” regarding efficiency of nitrogen application. We would appreciate if the authors could provide a reference where the concept of “economic optimum” is well explained.

3. Please provide full form and bibliographic reference of SSD (line 158). Please also mention in the text and provide reference to the “National blanket recommendation” as mentioned in response to comment number 5.

Reference:

Mattigod SV, Frampton JA, Lim CH (1985) Effect of ion-pair formation on boron adsorption by kaolinite. Clay Clay Miner 33:433–437

Reviewer #2: (No Response)

7. PLOS authors have the option to publish the peer review history of their article (what does this mean?). If published, this will include your full peer review and any attached files.

Reviewer #1: **Yes: **PRAVAT UTPAL ACHARJEE

Reviewer #2: No

---

## [Author Response · Author response to Decision Letter 2]

5 Mar 2021

We appreciate constant and sincere effort of the authors in making the revised manuscript better. However, in case of some responses made by the authors, it is hard for us to agree. These are as follows-

1. Regarding the response of point 10, we would like to mention that we have never suggested the term “nutrition” to be ‘not suitable’ in our comment number 8. Instead we mentioned that we would prefer if the authors include the terms “nutrient response” and “yield response” into the title. We urge the authors to be cautious on making such acquisitions.

[Response]: We would like to thank reviewer for their insightful review remarks. Please accept our sincere apologies for the inconvenience. We tried our best to address your concern in our manuscript. After careful and critical discussion among the authors, we have decided to change the title of the manuscript as per the reviewer’s suggestion. Here is the sequence of the manuscript title that we have changed with the progress of the review. 

Original title: Revised potassium nutrition in rice-wheat cropping system of Nepal

After first revision: Revision of potassium rate in the rice-wheat cropping system in different agro-ecozones of Nepal.

After second revision: Potassium nutrient response in the rice-wheat cropping system in different agro-ecozones of Nepal.

So, we are grateful with the reviewer to re-shape our manuscript title that is justifiable to our work.

We understand the author’s concern that a revised recommendation will help in making accurate policies in the country. However, as mentioned earlier, it is not only unscientific, but also unethical to declare dose/s of potassium as revised for the entire country based on experimental research in only 3 research stations, without considering being crossed checked the results in farmer’s field even for once! Rigorous field test is a must for any recommendation. 

[Response]: We now believe the revised manuscript title will address this concern of reviewer. We also have changed our argument in abstract and introduction to justify the manuscript title. We have removed the “a new fertilizer recommend rate” from the recommendation section of abstract (line 44), however, suggest to increase new K rate that and suggest to intensify this work in different locations (line 47-48). Hope this will address the reviewer’s concern in the right way.

Furthermore, Potassium, being an active cation, also influences dynamics of other nutrient ions in soil as well as in plant body. For example, Potassium indirectly influences B availability due to its effect on Ca absorption (Mattigod et al. 1985) and it is well known that a very narrow range of deficiency and toxicity of boron in plant body can severely affect the yield. This is just one example… there could be many more! 

[Response]: Yes, we agree our data limits us to explain this influence. We have not witnessed any of such deficiency and toxicity symptoms in the field crop in any of the research plots assuming that they are not likely to influence our results. 

As mentioned in response of point 7, the authors have provided no robust explanations in discussions of their own findings (like why there is a 30-33% rice yield decline in Rampur site in 2nd year? Why there is a 38% grain yield and 20% straw yield decline over 3 years in Rampur site? Etc…). And as the researchers did not check for the dynamics and uptake of potassium content in plant parts, there is no direct evidence of how potassium influenced the yield in this experiment. The limitation of this work itself inhibits from calling it a ‘Revision’. 

[Response]: Overall, we see the yield increment with increasing potassium rate, so we focused our discussion on this point. We respect the reviewer’s concern. This decline is not following in the next year and the yield response is increasing with increased K rate. So, we concentrated discussing our point highlighting the importance of K increment for increased yield. 

Yes, we accept our limitation of our study and the discussion points that we have incorporated in previous revised version (recall: line 403-405). After changing the title and hypothesis after this third revision we think this will justify the current motivation of the study.

Therefore, we could not be more positive here for the publication of this article due to aforesaid reasons and absence of the author’s effort in modifying these key concerns even after two revision attempts and we would like to rely upon the respected Editors for the final decision.

[Response]: We think we have addressed the concern of the reviewer by changing the title and motivation of our work along with changing the conclusion and last paragraph of the discussion section. We completely change the idea of ‘new K rate recommendation’ to ‘response of K fertilizer application’ that makes the whole essence of this manuscript in a way more logical than previous version. We will always be thankful to the reviewers for their insightful comments and suggestions. 

We came up with a very long way to make this manuscript to this publication stage. We believe the result and recommendation of this paper will be an important testimony to make national K fertilizer management plan and provides a strong support for conducting further field verification trials. So, we are very much hopeful with reviewer and editor to consider this revised manuscript for the publication. 

We really appreciate and would like to thank the authors for the other necessary modifications made so far. Although this manuscript has some merit regarding positive responses of potassium application on rice and wheat; the authors should be more considerate in segregating research hypothesis in context with the unexplained portions from result and discussion segment and need to reconstruct the title according to the objectives achieved through this research. 

[Response]: We have reconstructed the title, objective and hypothesis (last paragraph of the introduction, line 98-104) of our research which address the limitation of our work raised by reviewer. We would like to thank reviewer for this great suggestion. 

Some more points that could be taken under consideration in future are as follows-

2. Bibliographic reference of “Economic optimum” as provided by the authors is not robust enough to understand the application of this procedure in the present context. We have downloaded and read the book which includes some implications of “economic optimum” regarding efficiency of nitrogen application. We would appreciate if the authors could provide a reference where the concept of “economic optimum” is well explained.

[Response]: We have added the reference of the ‘economic optimum’ concept in the revised manuscript (line 204).

3. Please provide full form and bibliographic reference of SSD (line 158). Please also mention in the text and provide reference to the “National blanket recommendation” as mentioned in response to comment number 5.

[Response]: We have added the details of bibliographic reference as [36] in line 163.

We also have added the reference of national blanket fertilizer recommendation (line 181).

Reference:

Mattigod SV, Frampton JA, Lim CH (1985) Effect of ion-pair formation on boron adsorption by kaolinite. Clay Clay Miner 33:433–437

---

## [Editor Report · Decision Letter 3]

8 Mar 2021

Potassium nutrient response in the rice-wheat cropping system in different agro-ecozones of Nepal

PONE-D-20-12836R3

Dear Dr. Ojha,

We’re pleased to inform you that your manuscript has been judged scientifically suitable for publication and will be formally accepted for publication once it meets all outstanding technical requirements.

Kind regards,

Vassilis G. Aschonitis

Academic Editor

PLOS ONE
---

## [Editor Report · Acceptance letter]

10 Mar 2021

PONE-D-20-12836R3 

Potassium nutrient response in the rice-wheat cropping system in different agro-ecozones of Nepal 

Dear Dr. Ojha:

I'm pleased to inform you that your manuscript has been deemed suitable for publication in PLOS ONE. Congratulations! Your manuscript is now with our production department. 

Kind regards, 

on behalf of

Dr. Vassilis G. Aschonitis 

Academic Editor

PLOS ONE